# Feature out! Let Raw Image as Your Condition for Blind Face Restoration

**Xinmin Qiu** [* 1]   **Gege Chen** [* 1]   **Bonan Li** [1]   **Congying Han** [1]   **Tiande Guo** [1]   **Zicheng Zhang** [1]

## Abstract

Blind face restoration (BFR), which involves converting low-quality (LQ) images into high-quality (HQ) images, remains challenging due to complex and unknown degradations. While previous diffusion-based methods utilize feature extractors from LQ images as guidance, using raw LQ images directly as the starting point for the reverse diffusion process offers a theoretically optimal solution. In this work, we propose ***Pseudo-Hashing Image-to-image Schrödinger Bridge (P-I2SB)***, a novel framework inspired by optimal mass transport problems, which enhances the restoration potential of Schrödinger Bridge (SB) by correcting data distributions and effectively learning the optimal transport path between any two data distributions. Notably, we theoretically explore and identify that existing methods are limited by the optimality and reversibility of solutions in SB, leading to suboptimal performance. Our approach involves preprocessing HQ images during training by hashing them into pseudo-samples according to a rule related to LQ images, ensuring structural similarity in distribution. This guarantees optimal and reversible solutions in SB, enabling the inference process to learn effectively and allowing P-I2SB to achieve state-of-the-art results in BFR, with more natural textures and retained inference speed compared to previous methods.

## 1. Introduction

Blind face restoration (BFR) (Yang et al., 2021; Wang et al., 2021; Chen et al., 2021a) presents a significant challenge in the field of image restoration due to the complexity of degradation and the unknown causes of such degradation, along with high demands for the quality of restoration results. As a blind inverse problem, BFR necessitates the incorporation of extensive prior information during the restoration process. In recent years, generative models (Ho et al., 2020; Rombach et al., 2022; Goodfellow et al., 2014; Liu et al., 2022; Karras et al., 2022; Li et al., 2025) based on diffusion processes have demonstrated promising results in BFR. These models have evolved from focusing on the transport between Gaussian noise and real data distributions to facilitating transport between arbitrary real data distributions. In contrast, bridge-based methods address the transport paths between arbitrary distributions. An increasing number of image processing tasks in the visual domain, including image inpainting, and image segmentation, are capitalizing on this capability. Diffusion-based generative approaches offer robust data priors and a comprehensive solution framework. Particularly in image restoration tasks, which require transforming low-quality (LQ) images into high-quality (HQ) counterparts with clear details and consistent content.

Recent strategies to address these issues can be categorized into two primary approaches. The first approach involves utilizing non-diffusion-based methods for preliminary processing, followed by refinement through diffusion-based techniques. Examples include DifFace (Yue & Loy, 2022) and DR2 (Yue & Loy, 2022), which employ CNN-based models like SwiIR (Liang et al., 2021) for initial restoration, enhancing texture clarity using diffusion-based methods. The second approach involves designing sophisticated feature extraction and fusion techniques to refine condition guidance. Methods such as DiffBIR (Lin et al., 2023) and PGDiff (Yang et al., 2024) incorporate external modules to improve the integration of LQ images as condition guidance, which indirectly controls the inverse process. However, these approaches introduce additional complexities. The preliminary models are constrained by their lightweight nature, potentially leading to the loss of critical information. Meanwhile, the latter requires meticulously designed modules to effectively guide the path, which can weaken the inherent transport capabilities of diffusion models.

Both these methods share a focus on learning generative models for HQ images by using feature extractors and fusion to derive and embed content features from LQ images into the model as guidance conditions. However, the forward and reverse processes of diffusion models are not inherently linked to LQ images, limiting restoration performance due to the constraints of LQ feature extraction. Thus, robust feature

---

[*]Equal contribution  [1]School of Mathematical Sciences, University of Chinese Academy of Sciences, Beijing. Correspondence to: Congying Han \<hancy@ucas.ac.cn\>.

*Proceedings of the $42^{nd}$ International Conference on Machine Learning*, Vancouver, Canada. PMLR 267, 2025. Copyright 2025 by the author(s).

extraction and embedding for LQ images are essential to fully exploit diffusion models, such as DDPM (Ho et al., 2020; Nichol & Dhariwal, 2021), which leverage learned transformations between Gaussian noise and HQ images. Consequently, we explore the direct construction of optimal transport paths directly between LQ and HQ images, where Schrödinger Bridge (SB) based methods can effectively achieve this goal.

In this work, we propose P-I2SB, a novel restoration framework, which consists of two stages: the **Pseudo-Hashing Module** (PHM) for preprocessing image pairs using the pseudo-hashing strategy, and the **Schrödinger Bridge Module** (SBM) for directly finding the optimal path between two data distributions without the need for indirect guidance through Gaussian noise. In the **second stage** (SBM), we employ I2SB (Liu et al., 2023), a robust restoration framework based on SB. I2SB leverages the forward and backward stochastic differential equation models inherent in SB, connecting with stochastic gradient models (SGM) through conditional configurations. While effective in single-degradation image restoration tasks, the performance in blind tasks is suboptimal. We address this by examining how the optimal transport problem in image restoration can establish the forward and backward stochastic differential equation models of SB. Notably, blind tasks do not conform to the typical Monge optimal transport problem, where the *optimal transport* between images is not unique. A single high-quality image may correspond to multiple optimal paths from various LQ images, each facilitating restoration back to the high-quality image. This contradicts the optimality principle of reversible SB, hindering the development of a consistent I2SB model. We explore this theoretical foundation, as illustrated in Figure 1, and introduce a novel pseudo-hashing preprocessing strategy, which constitutes the **first stage** (PHM) of our method. This technique processes training image pairs in blind tasks by transforming the associations between high-quality and low-quality distributions, ensuring the transformation is reversible.

Instead of relying on external methods or designing complex networks, our work *focuses on correcting unreasonable endpoint constraints to fully exploit the potential of SB models for learning optimal transport paths compared to previous*. We identify that conflicts arise from the forward process definition in Vanilla-I2SB theoretically, where a single HQ image can correspond to multiple degraded versions, resulting in divergent forward processes. The core of our research is the exploration of these improvements from both theoretical and experimental perspectives, demonstrating their effectiveness, which confirms the practical efficiency and economic viability of the proposed enhancements. Compared to guidance-based methods, our approach achieves state-of-the-art results within the same inference time, particularly on datasets with complex degradations, outperform-

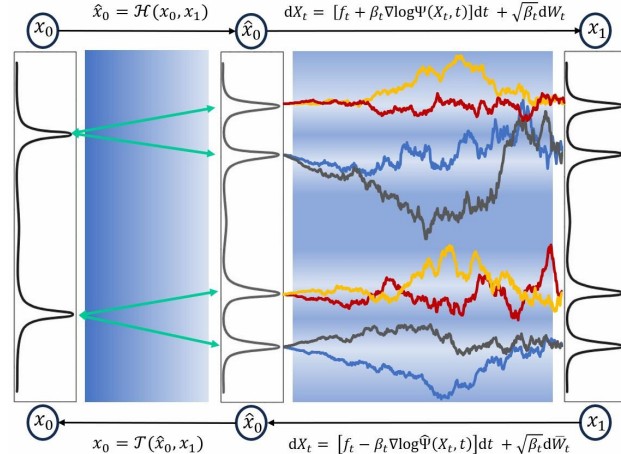

*Figure 1.* **Theoretical Foundations**. We first pseudo-hashing $x_0$ to $\hat{x}_0$ using the information about $x_1$ so that $\hat{x}_0$ has a similar distribution structure with $x_1$, and then build Schrödinger Bridge Module (SBM) between $\hat{x}_0$ and $x_1$. Both two processes are reversible.

ing previous diffusion-based and flow-based methods. By directly identifying optimal transport paths between LQ and HQ distributions, our framework aligns more closely with the nature of reverse problems, achieving superior results with natural and clear textures and reduced distortions.

## 2. Related Work

### 2.1. Blind Face Restoration

Blind face restoration (Li et al., 2020; Wang et al., 2021; 2022; Qiu et al., 2023; 2024) is a crucial subfield within image restoration. Significant advances have been proposed, including various methodological approaches such as CNN-based (Menon et al., 2020; Li et al., 2020), GAN-based (Yang et al., 2021; Wang et al., 2021; Chen et al., 2021a), dictionary-based (Gu et al., 2022; Zhou et al., 2022; Wang et al., 2022), and diffusion-based (Yue & Loy, 2022; Wang et al., 2023) methods. GFPGAN (Wang et al., 2021) and GPEN (Yang et al., 2021) leverage GAN-based generative models within an encoder-decoder framework to embed face prior. RestoreFormer (Wang et al., 2022) integrates classical dictionary-based methods with contemporary vector quantization (VQ) techniques (Esser et al., 2021). DiffBIR (Lin et al., 2023) enhances prior knowledge by employing pre-trained stable diffusion as generation priors. FlowIE (Zhu et al., 2024) and PMRF (Ohayon et al., 2024) are image enhancement methods based on rectified flow (Liu et al., 2022).

### 2.2. Schrödinger Bridge in Diffusion

Schrödinger Bridge (SB) problem (Schrödinger, 1932; Léonard, 2013; Chen et al., 2016) represents an entropy-regularized formulation of optimal transport. Given two arbitrary probability distributions, $p(x_0)$ and $p(x_1)$, the SB

framework seeks to determine an optimal coupling path between these distributions. To address this challenge, Denoising Diffusion Bridge Models (DDBMs) (Zhou et al., 2023) have been introduced, which leverage the concept of driving the diffusion process to establish a diffusion bridge. Image-to-Image Schrödinger Bridge (I2SB) (Liu et al., 2023) adopts a computational framework analogous to standard score-based models. This approach has demonstrated exceptional performance across a variety of image-to-image translation tasks. Unlike conventional diffusion models (Ho et al., 2020; Nichol & Dhariwal, 2021; Song et al., 2020; Rombach et al., 2022), I2SB constraints necessitate careful consideration to ensure optimal performance in practical applications.

## 3. Preliminaries

We begin by establishing some notation and providing an overview of the problem. From a Bayesian perspective (Davison, 2003; Kaipio & Somersalo, 2006), any image $x \in \mathbb{R}^d$, whether perceived as clear or degraded, can be viewed as a realization of a random variable $X$. We consider two probability distributions, denoted as $\nu_0$ and $\nu_1$, which correspond to the measures of the spaces of HQ and LQ images, respectively. These distributions are expressed as $p_0(x)dx = \nu_0(dx)$ and $p_1(y)dy = \nu_1(dy)$. Given a LQ image $y$ sampled from $p_1(y)$, the objective of an image restoration algorithm is to generate a prediction $\widetilde{x}$ that closely approximates the corresponding clear image $x \sim p_0(x)$.

### 3.1. Optimal Mass Transport Problem

The image restoration problem can be viewed as a type of optimal mass transport (OMT) problem. To begin, we consider the general framework of optimal mass transport, specifically Monge's OMT problem (Chen et al., 2021b). Let $\mathcal{T}_{\nu_0\nu_1} := \{T : \mathbb{R}^d \to \mathbb{R}^d \mid T\#\nu_0 = \nu_1\}$ represent a family of measure-preserving maps, where $T\#\nu_0 = \nu_1$ indicates that $\nu_1$ is the push-forward of $\nu_0$ under the map $T$. Additionally, let $c : \mathbb{R}^d \times \mathbb{R}^d \to [0, +\infty)$ be a cost function that quantifies the cost of transporting a unit of mass from a location $x \sim p_0(x)$ to a location $y \sim p_1(y)$.

$$\inf_{T \in \mathcal{T}_{\nu_0\nu_1}} \int_{\mathbb{R}^d} c(x, T(x))\nu_0(dx). \tag{1}$$

If the map $T$ is reversible, then for a given $y \sim p_1(y)$, we have $x = T^{-1}(y) \sim p_0(x)$ as the restored image of $y$ in the context of image restoration. However, Monge's problem does not always admit a solution. For instance, consider the case where $\nu_0$ is a Dirac distribution and $\nu_1$ is the sum of two Dirac distributions, each with half the magnitude. Since $\nu_0$ must be "split" to be transported to two distinct locations, no map exists between them, rendering $\mathcal{T}_{\nu_0\nu_1}$ an empty set.

Therefore, the more extensively studied OMT problem is a **relaxed** version introduced by Kantorovich in the 1940s[1]. This problem seeks an optimal joint distribution within a non-empty joint distribution space. Its dynamic formulation was elegantly developed by Benamou and Brenier in (Benamou & Brenier, 2000), and is expressed as:

$$\inf_{(\mu,v)} \int_0^1 \int_{\mathbb{R}^d} ||v(t,x)||^2 \mu_t(dx)dt, \\ \frac{\partial\mu}{\partial t} + \nabla \cdot (v\mu) = 0, \quad \mu_0 = \nu_0, \mu_1 = \nu_1. \tag{2}$$

Here, the flow $\{\mu_t; 0 \le t \le 1\}$ represents a family of continuous mappings from the interval $[0, 1]$ to the set of probability measures on $\mathbb{R}^d$, each possessing a finite second moment. The variable $v$ denotes smooth vector fields associated with these mappings.

### 3.2. Schrödinger Bridge Problem

Let $\Omega = C([0, 1], \mathbb{R}^d)$ denote the space of all continuous $\mathbb{R}^d$-valued paths over the unit time interval $[0, 1]$. The sets $M_+(\Omega)$ and $P(\Omega)$ represent all positive measures and probability measures on the space $\Omega$, respectively. Given a reference path measure $R \in M_+(\Omega)$, the Schrödinger bridge problem (Schrödinger, 1932; Léonard, 2013; Chen et al., 2016) is formulated as:

$$\text{minimize} \quad H(P|R) = \mathbb{E}_P[\ln \frac{dP}{dR}], \\ \text{s.t.} \quad P \in P(\Omega) : P_0 = \nu_0, P_1 = \nu_1, \tag{3}$$

where $\nu_0, \nu_1 \in P(\mathbb{R}^d)$ represent the prescribed initial and final marginal distributions. The term $H(P \mid R)$ denotes the relative entropy of the measure $P$ with respect to the reference measure $R$. This problem admits at most one solution, denoted as $\hat{P} \in P(\Omega)$, which is referred to as the Schrödinger bridge from $\nu_0$ to $\nu_1$ over the measure $R$.

In general, we consider a reference family denoted as $\mathbb{D} \subset M_+(\Omega)$, which consists of Markov finite energy diffusion processes possessing various practical properties. The coordinate process $X_t(\omega) = \omega(t)$ can be represented:

$$dX_t = f_t(X_t)dt + \sqrt{\beta_t}dW_t, \\ dX_t = [f_t(X_t) - \beta_t \nabla \log p(X_t, t)]dt + \sqrt{\beta_t}d\bar{W}_t, \tag{4}$$

where $\beta_t \in \mathbb{R}$ represents diffusion coefficients, and $f_t : (\mathbb{R}^d, [0, 1]) \to \mathbb{R}^d$ denotes drift function. The term $\bar{p}(x, t)$ is the time-marginal density function of $X_t$ at time $t$. $W_t$ is the Wiener process, and $\bar{W}_t$ is its reverse process.

Given $R \in \mathbb{D}$, we can obtain a specific Schrödinger bridge $P^* \in \mathbb{D}$ from $\nu_0$ to $\nu_1$ over $R$, whose coordinate process

---

[1]More details can be seen in Appendix B

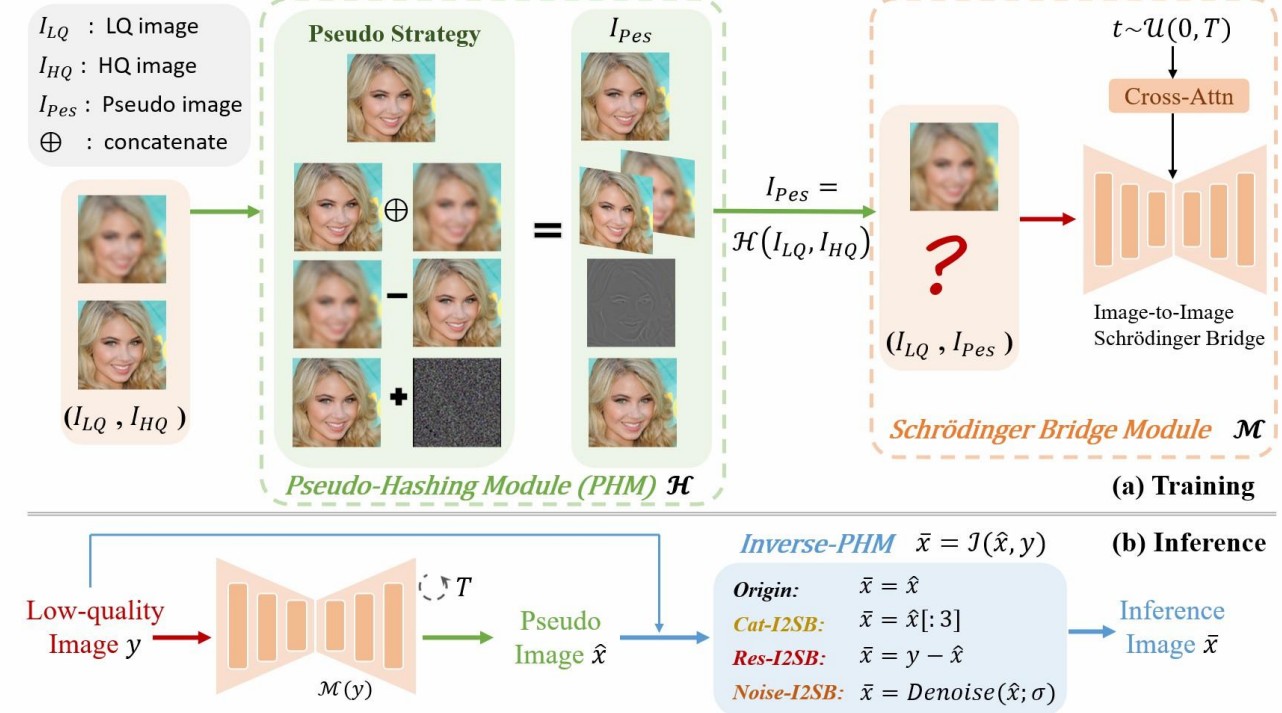

*Figure 2.* **Framework of the proposed P-I2SB** for blind face restoration task. P-I2SB is a bridge-based diffusion model incorporating a pseudo-hashing strategy. During training (a), given LQ and HQ face pairs, a Pseudo image is generated via the Pseudo-Hashing Module (PHM), including the strategies of Origin, Cat-I2SB, Res-I2SB, and Noise-I2SB. In inference (b), the model produces a Pseudo Image from an LQ input, which is then converted to a high-quality result using the inverse pseudo-hashing method.

$X_t^*$ can be represented as follows (Chen et al., 2023):

$$dX_t = [f_t + \beta_t \nabla \log \Psi(X_t, t)]dt + \sqrt{\beta_t}dW_t,$$
$$dX_t = [f_t - \beta_t \nabla \log \hat{\Psi}(X_t, t)]dt + \sqrt{\beta_t}d\bar{W}_t, \quad (5)$$

$$\text{s.t.} \quad X_0 \sim p_0, X_1 \sim p_1. \quad (6)$$

The functions $\Psi, \hat{\Psi} \in C^{2,1}(\mathbb{R}^d, [0,1])$ represent time-varying energy potentials that solve a set of known coupled partial differential equations (PDEs).

The density function $p^*(t, x)$ of $X_t^*$ serves as the solution to the following optimal transport problem (OTM) (Chen et al., 2021b):

$$\inf_{(p,v)} \int_0^1 \int_{\mathbb{R}^d} \left(\frac{1}{2}||v(t,x) - \bar{v}(t,x)||^2 + \right.$$
$$\left. \frac{\beta_t}{8}||\nabla \log \frac{p(t,x)}{\bar{p}(t,x)}||^2\right)p(t,x)dxdt, \quad (7)$$

$$\text{s.t.} \quad \frac{\partial p}{\partial t} + \nabla \cdot (vp) = 0,$$
$$p(0, \cdot) = p_0(\cdot), \quad p(1, \cdot) = p_1(\cdot), \quad (8)$$

where $\bar{v}(t,x) := f_t(x) - \frac{\sqrt{\beta_t}}{2}\nabla \log \bar{p}(t,x)$ and $p(t,x)dx = \mu_t(dx)$, just similar as equation (2).

## 4. Methodology

Here, we introduce the Pseudo-Hashing-I2SB (P-I2SB). We begin by theoretically exploring how SB retrieves clearer HQ images, presenting a theorem to justify the rationale of I2SB and demonstrating its limitations for BFR through Corollary (Sec. 4.1). To resolve this issue, we propose the Pseudo-Hashing Module (PHM) with three implementations: Res-I2SB, Cat-I2SB, and Noise-I2SB (Sec. 4.2), alongside the Schrödinger Bridge Module (SBM). Additionally, we perform a toy experiment on the MNIST dataset for quick verification (Sec. 4.3).

### 4.1. Practical Application and Improvement of SB

Building on the previous discussion, we derive an optimal reversible Markov diffusion process $X_t^*$ that describes the evolution of the image state from clear to degraded and vice versa between two boundary distributions. In practice, the explicit forms of $p_0$ and $p_1$ are unknown. Instead, we have access to sample pairs $\{(x_i, y_i) \mid x_i \sim p_0, y_i \sim p_1\}_{i=1}^N$, where $x_i$ denotes a clear image and $y_i$ its corresponding degraded counterpart.

For image processing tasks, a practical approach involves learning the reverse process from the forward process of a self-designed reversible Markov diffusion process $X_t$. As for image restoration, we posit that $X_t$ is optimal when its

forward process transforms a clear image into its corresponding degraded counterpart, which is a stringent condition.

This leads us to consider the following question: given a stochastic process defined over the time interval $(0, 1)$:

$$\mathrm{d}X_t = \alpha_t(X_t)\mathrm{d}t + \sqrt{\beta_t}\mathrm{d}W_t. \tag{9}$$

To ensure that the process $X_t^{xy}$ can be described by Equation (9) for $t \in (0, 1)$, it is necessary to appropriately select time boundary distributions. $X_t^{xy}$ represents the stochastic process subject to the conditions $X_0 = x$ and $X_1 = y$.

Generally, for any positive measure $Q \in \mathrm{M}_+(\Omega)$ on $\Omega$, its disintegration formula is expressed (Léonard, 2013):

$$Q(\cdot) = \int_{\mathbb{R}^{2d}} Q^{xy}(\cdot)Q_{01}(\mathrm{d}x\mathrm{d}y), \tag{10}$$

where $Q_{01} = (X_0, X_1)_\# Q \in \mathrm{M}_+(\mathbb{R}^{2d})$, and $Q^{xy} = Q(\cdot \mid X_0 = x, X_1 = y) \in \mathrm{P}(\Omega)$, referred to as the bridge of $Q$ between the pairs $(x, y)$.

**Theorem 4.1.** *For $P \in \mathbb{D}$ whose coordinate process $X_t$ is the solution of the boundary value problem (5,6), then the coordinate process $X_t^{xy}$ of $P^{xy}$ is also subject to the equation (5) with boundary condition $X_0 = x, X_1 = y$ if $p_0(\cdot) = \delta_x(\cdot)$ centered at $x \in \mathbb{R}^d$ and $p_1(\cdot) = \delta_y(\cdot)$ centered at $y \in \mathbb{R}^d$.*

Theorem 4.1 indicates that an identical stochastic process over the time interval $(0, 1)$ can be learned using a large number of sample pairs, which are assumed to follow Dirac delta distributions.

Take I2SB for example, which effectively utilizes the SGM framework to construct SB for specific known degradation types. However, complexities arise in BFR, such as a single HQ image corresponding to multiple types of LQ images. In this scenario, all LQ images are assumed to follow a certain distribution, resulting in complex sample pairs like $\{(x_i, y_{i_j}) \mid x_i \sim p_0, y_{i_j} \sim p_1\}, (i = 1, \ldots, N, j = 1, \ldots, M)$. Despite this, constructing a unified SB, as anticipated by I2SB, is not feasible. The underlying reason is derived directly from Theorem 4.1, which we present as a corollary:

**Corollary 4.2.** *If we have two pairs like $(x, y_1)$ and $(x, y_2)$, then we can not build a unified Schrödinger bridge $P \in \mathbb{D}$ between them by means shown in theorem 4.1.*

This follows from the uniqueness of the solution to a stochastic differential equation given an initial value.

### 4.2. Pseudo-Hashing-I2SB

***Pseudo-Hashing Module*** Initially, for the given high-quality (HQ) and low-quality (LQ) image pairs, we propose the **Pseudo-Hashing Module (PHM)** as a preprocess-

---

**Algorithm 1** Training

1: **Input:** clean $p_\mathcal{A}(\cdot)$ and degraded $p_\mathcal{B}(\cdot|x)$
2: **repeat**
3:   $t \sim \mathcal{U}, x \sim p_\mathcal{A}(X_0), y \sim p_\mathcal{B}(X_1|X_0)$
4:   $\hat{x} = \mathcal{H}(x, y)$, according to (11)
5:   $X_t \sim q(X_t|X_0, X_1)$, according to (15)
6:   $\mathcal{L} = w_t \parallel \epsilon(X_t, t; \theta) - \frac{X_t - X_0}{\sigma_t} \parallel$, take SGM
7: **until** converges

---

**Algorithm 2** Inference

1: **Input:** degraded $X_N \sim p_\mathcal{B}$, trained $\epsilon(\cdot, \cdot; \theta)$, Step $N$
2: **for** $n = N$ **to** $1$ **do**
3:   Predict $X_0^\epsilon$ using $\epsilon(X_n, t_n; \theta), t_n = \frac{n}{N}$
4:   $X_{t_{n-1}} \sim p(X_{n-1}|X_0^\epsilon, X_n)$, according to (16)
5: **end for**
6: $\bar{x} = \mathcal{J}(X_0, X_N)$, according to (18)
7: **return:** $\bar{x}$

---

ing to address the infeasibility issue highlighted in Corollary 4.2. The unified training process is detailed in Algorithm 1, where $x$ and $y$ represent HQ and LQ images, respectively. Specifically, considering the feasibility of image pair configurations, we propose three detailed pseudo-hashing strategies within PHM, as illustrated in Formula (11).

$$\hat{x} = \mathcal{H}(x, y) = \begin{cases} x, & \text{Origin,} \\ x \oplus y, & \text{Cat-I2SB,} \\ x - y, & \text{Res-I2SB,} \\ x + \sigma(y)\epsilon, & \text{Noise-I2SB.} \end{cases} \tag{11}$$

The first strategy is sample concatenation, referred to as **Cat-I2SB**. Given a high-quality image $I_{hq}$ and its corresponding randomly degraded LQ image $I_{lq}^d = \text{Degradation}_d(I_{hq})$, where $d$ denotes the random degradation type and parameters, we perform channel concatenation. Specifically, $I_{hq}$ and $I_{lq}^d$ are concatenated to form a new image, while $I_{lq}^d$ is concatenated with itself. Consequently, the training image pair becomes:

$$(\hat{x}, y) = (I_{hq} \oplus I_{lq}^d, I_{lq}^d \oplus I_{lq}^d). \tag{12}$$

The second strategy is relative residual prediction, termed **Res-I2SB**. From a spatial geometric perspective, the direction of $R^d = I_{lq}^d - I_{hq}$ aligns with $I_{hq}$, as illustrated in Figure 3(a). By setting the two endpoint distributions of the image pair as:

$$(\hat{x}, y) = (R^d = I_{lq}^d - I_{hq}, I_{lq}). \tag{13}$$

We achieve the goal of hashing the endpoint distributions while maintaining a consistent direction and distance between the hashed endpoints.

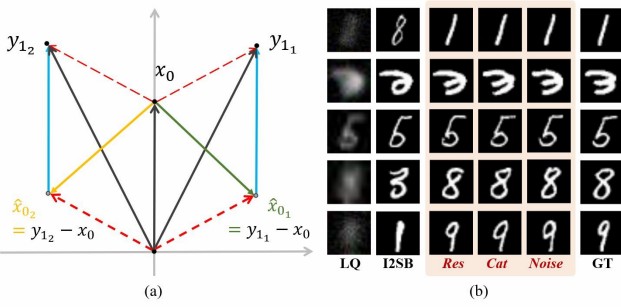

(a)         (b)

*Figure 3.* **Toy Experiments**. (a) illustrates Pseudo-hashing of $x_0$ in Res-I2SB within the $\mathbb{R}^d$ space. For a single HQ image $x_0$ with two LQ counterparts, $y_{1_1}$ and $y_{1_2}$, $x_0$ is split into $\hat{x}_{0_1}$ and $\hat{x}_{0_2}$ using residuals between $(x_0, y_{1_1})$ and $(x_0, y_{1_2})$. This allows the construction of a unified SBM, depicted by blue arrows between $(\hat{x}_{0_1}, y_{1_1})$ and $(\hat{x}_{0_2}, y_{1_2})$. (b) presents results of toy experiments.

The third strategy involves lightly adding noise, referred to as **Noise-I2SB**. A small amount of Gaussian noise is added to $I_{hq}$, which does not cause significant visual degradation but directly alters the distribution of the image vector $I_{hq}$. This is expressed as $I_{hq}^{noise} = I_{lq}^d + \lambda_d \epsilon$, where $\lambda_d$ represents the noise level, directly related to $I_{lq}^d$. The image pair is then set as:

$$(\hat{x}, y) = (I_{hq}^{noise}, I_{lq}). \tag{14}$$

This method leverages the most common Gaussian noise correction in SGM, applying minimal noise to the image to redistribute its numerical components into a new distribution without compromising image content, as shown in Figure 2.

***Schrödinger Bridge Module***    After preprocessing in the Pseudo-Hashing Module (PHM), we obtain the image pair $(\hat{x}, y)$, which is used to design the **Schrödinger Bridge Module (SBM)** for training these predefined image pairs. These pairs are treated as two new data distributions for SBM to learn. Inspired by I2SB (Liu et al., 2023), we directly employ the forward and reverse processes of I2SB. The forward process is defined as follows:

$$X_t \sim q(X_t|X_0, X_1) = \mathcal{N}(X_t; \mu_t(X_0, X_1), \Sigma_t),$$

$$\mu_t = \frac{\bar{\sigma}_t^2}{\bar{\sigma}_t^2 + \sigma_t^2}X_0 + \frac{\sigma_t^2}{\bar{\sigma}_t^2 + \sigma_t^2}X_1, \Sigma_t = \frac{\sigma_t^2 \bar{\sigma}_t^2}{\bar{\sigma}_t^2 + \sigma_t^2} \cdot I \tag{15}$$

where $\bar{\sigma}_t$ and $\bar{\sigma}_t$ is the scheduled parameter. The reverse process follows the design of the SGM framework, as shown:

$$X_{t-1} \sim p(X_{t-1}|X_0^\epsilon, X_t), \tag{16}$$

where $\epsilon$ denotes the trained model using a Unet network. Additionally, during training, we incorporate the loss as follows:

$$\mathcal{L} = \| \epsilon(X_t, t; \theta) - \frac{X_t - X_0}{\sigma_t} \|, t \sim \mathcal{U}(0, T), \tag{17}$$

where the analysis of losses is detailed in Appendix F.

Correspondingly, the results of the pseudo-hashing need to be restored in the inference process to obtain the corresponding high-quality image of the final repair. We give a unified restoration process:

$$\bar{x} = \mathcal{J}(\hat{x}, y) = \begin{cases} \hat{x}, & \text{Origin,} \\ \hat{x}[:3], & \text{Cat-I2SB,} \\ \hat{x} - y, & \text{Res-I2SB,} \\ Denoise(\hat{x}; \sigma(y)), & \text{Noise-I2SB.} \end{cases} \tag{18}$$

where *Denoise* uses DDIM (Song et al., 2020) at the initial number of steps to denoise the image. Consequently, we obtain the final clear restoration result, denoted as $\bar{x}$. The inference process is detailed in Algorithm 2.

### 4.3. Toy Exploration and Analysis

To directly investigate the impact of various hashing strategies within I2SB, we conducted comparative experiments on the MNIST dataset. The MNIST dataset consists of 28x28 grayscale images of handwritten digits. We denote HQ images as $I_{hq}$ and generate LQ images $I_{lq}^d$ by applying random manual degradation. Both training and inference processes utilize a lightweight model implemented at a resolution of 64x64. The comparative methods include Vanilla-I2SB and variations with different hashing strategies: Res-I2SB, Cat-I2SB, and Noise-I2SB. As depicted in Figure 3 (b), while Vanilla-I2SB and the three hashing strategies can all restore degraded images of handwritten digits to clear images, there are discrepancies in label classification accuracy and simple texture representation. This indicates that the improved methods preserve the structural integrity of image content more accurately during the restoration of LQ images.

We recognize a strong correlation between the task constraints of image-to-image translation and the data distributions. In previous methods like DDPM (Ho et al., 2020), LQ images are often used as guidance conditions to ensure accurate content inference. Techniques such as SR3 (Saharia et al., 2022) and IDM (Gao et al., 2023) enhance the embedding of LQ images to improve the ability of diffusion models to preserve image content during inference, which modifies the transmission path between noise and HQ images. We thoroughly explored the theoretical optimality of using the raw LQ image directly as one endpoint of diffusion process, instead of Gaussian distributions. In image-to-image transmission, directly altering the input form of endpoint images, rather than adding complex network structures for feature extraction, is more straightforward and efficient. Both theoretical and experimental results confirm that this approach effectively aids the SB in identifying the optimal path.

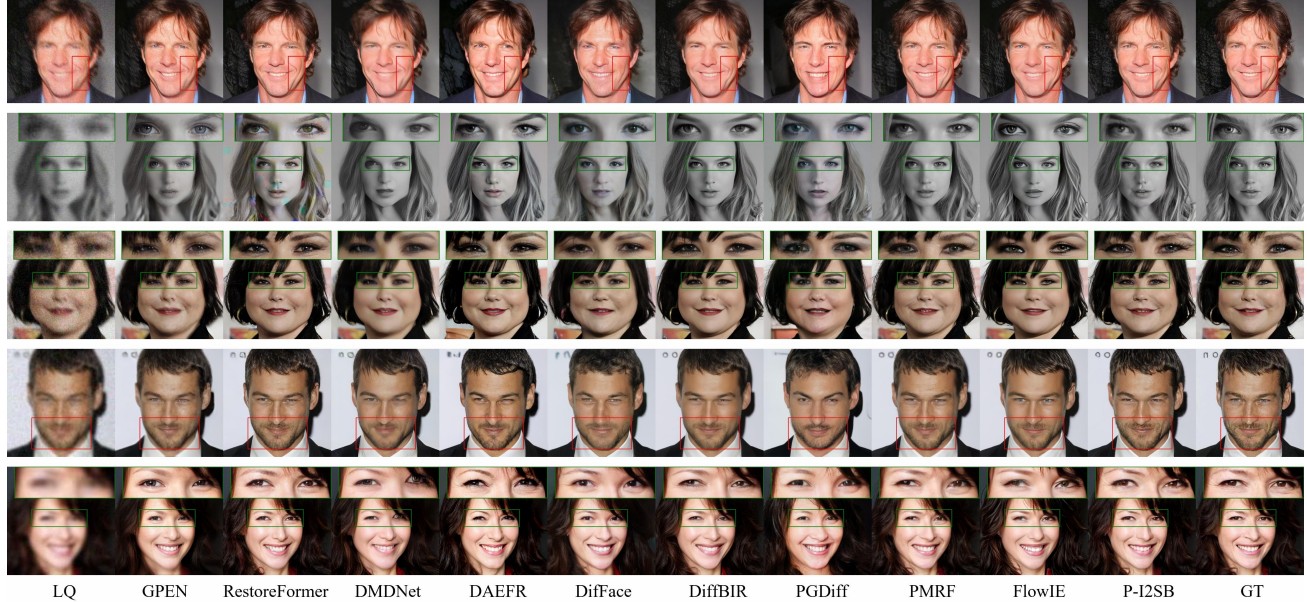

*Figure 4.* **Qualitative comparisons on the CelebA-Test** for blind face restoration. Taking Res-I2SB as an example, our P-I2SB demonstrates strong performance in detail enhancement, hue preservation and attitude preservation compared to these latest GAN-based, dictionary-based, and diffusion-based methods. **Zoom in for best view**.

## 5. Experiments

**Training Datasets**   We choose FFHQ (Karras et al., 2019) as training dataset, which contains 70,000 high-quality PNG format face images with $1024 \times 1024$ resolution. In this experiment, we resize all images to $512 \times 512$ for training.

**Preprocessing**   Since our P-I2SB is supervised training, the corresponding LQ-HQ image pairs are required. We use generated random degradation model to simulate LQ images in the real world. Its generation formula (Zhang et al., 2018a; Wang et al., 2021) is shown in Eq.(19), where $x$ is HQ image, $k_\sigma$ is Gaussian blur kernel, $r$ represents downsampling scale factor, and $q$ represents JPEG compression with quality factor $q$. The parameters $\sigma, r, \delta, q$ are randomly sampled from {0.1: 10}, {0.8: 8}, {0: 20}, {60: 100}, to align with the experimental environment of recent methods for BFR, as shown in Appendix E. We also add gray color probability during the training process for color adaptation and augment data with the horizontal flip.

$$y = [(x \otimes k_\sigma) \downarrow_r + n_\delta]_{\mathrm{JPEG}_q} \qquad (19)$$

**Testing Datasets**   We choose CelebA-Test, Real-World LFW, CelebChild and WedPhoto-Test as the testing datasets. CelebA-Test contains 3,000 HQ images randomly sampled from CelebA-HQ (Karras et al., 2018) with the resolution of $512 \times 512$. Similarly, the corresponding random LQ images are generated for evaluation by using the degradation model in Eq.(19) and the same set of parameters used in the training dataset. Real-World LFW, CelebChild and WedPhoto-Test are different real-world datasets to test the

generalization ability. All these datasets have no overlap with our training dataset.

### 5.1. Comparisons with State-of-the-art Methods

**Comparison Methods**   We compare Pseudo-Hashing with recent BFR methods, including PSFRGAN (Chen et al., 2021a), GFPGAN (Wang et al., 2021), GPEN (Yang et al., 2021), VQFR (Gu et al., 2022), CodeFormer (Zhou et al., 2022), RestoreFormer (Wang et al., 2022), DMDNet (Li et al., 2022), DAEFR (Tsai et al., 2023), DifFace (Yue & Loy, 2022), DR2 (Wang et al., 2023), PGDiff (Yang et al., 2024), DiffBIR (Lin et al., 2023), PMRF (Ohayon et al., 2024), FlowIE (Zhu et al., 2024) and I2SB (Liu et al., 2023).

**Metrics**   We quantitatively compare the differences between our method and SOTAs using five widely-used metrics, including SSIM (Wang et al., 2004), PSNR, FID (Heusel et al., 2017), NIQE (Mittal et al., 2012), and LPIPS (Zhang et al., 2018b). Among them, NIQE is a no-reference metric. SSIM and PSNR are pixel-wise similarity measures, while FID, NIQE and LPIPS are perceptual.

**Evaluation on Synthetic Dataset.**   We compared the restoration performance of P-I2SB, using Cat-I2SB as an example, with existing SOTA methods on synthetic CelebA-Test. Figure 4 shows visual comparison results, demonstrating that our approach can restore more accurate textures, colors, and local features in severely degraded facial images. VQFR and RestoreFormer exhibit noticeable color artifacts in their restoration outputs, while CodeFormer, DR2, PMFR, and FlowIE produce overly smooth results, losing local texture details. In contrast, our improved method achieves more

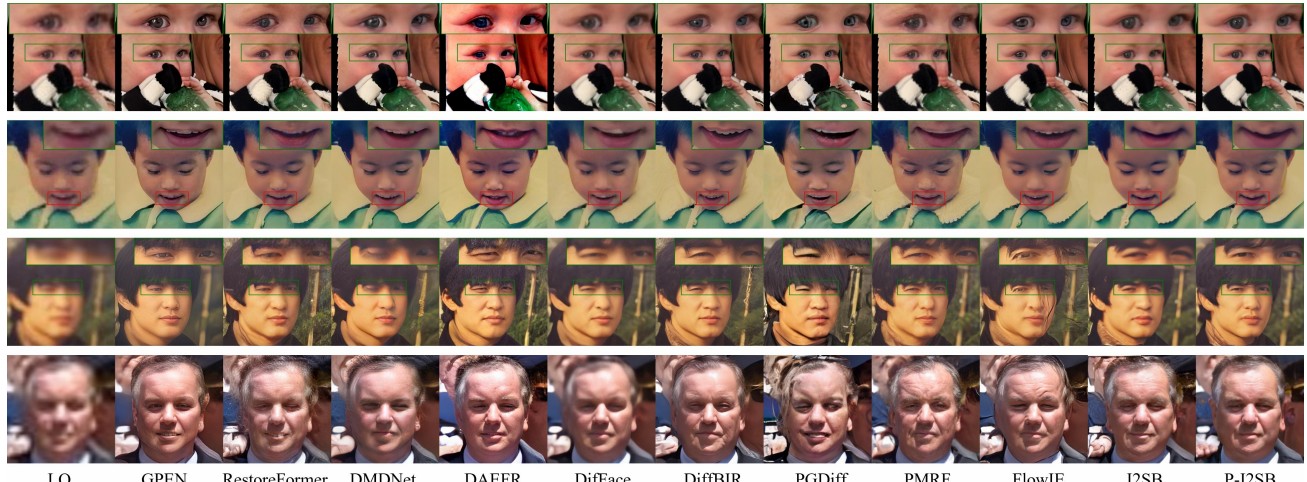

*Figure 5.* **Qualitative comparisons on real-world datasets**. Our P-I2SB demonstrates superior performance in both detail enhancement and hue preservation, particularly on inputs with severe degradation. **Zoom in for best view**.

*Table 1.* **Quatitative comparison on CelebA-Test** with 3000 randomly selected images for blind face restoration. **Bold** and underline indicate the best and the second best performance. Our P-I2SB excels in naturalness and perceptual metrics. GFP denotes GFPGAN and Restore denotes RestoreFormer. We compare with recent BFR methods, including GPEN (Yang et al., 2021), GFPGAN (Wang et al., 2021), RestoreFormer (Wang et al., 2022), DMDNet (Li et al., 2022), DAEFR (Tsai et al., 2023), DifFace (Yue & Loy, 2022), DiffBIR (Lin et al., 2023), DR2 (Wang et al., 2023), PGDiff (Yang et al., 2024), PMRF (Ohayon et al., 2024), FlowIE (Zhu et al., 2024) and I2SB (Liu et al., 2023).

| Metrics | Input | Methods | | | | | | | | | | | | P-I2SB |
| | | GPEN | GFP | Restore | DMDNet | DAEFR | DifFace | DiffBIR | DR2 | PGDiff | PMRF | FlowIE | I2SB | |
| | | *CVPR* | *CVPR* | *CVPR* | *TPAMI* | *ICLR* | *TPAMI* | *ECCV* | *CVPR* | *NIPS* | *ICML* | *CVPR* | *ICML* | |
| SSIM↑ | 0.6460 | 0.6777 | 0.6827 | 0.6219 | 0.6727 | 0.5892 | 0.6494 | 0.6570 | 0.6554 | 0.6220 | 0.6815 | 0.6479 | **0.7047** | 0.6581 |
| PSNR↑ | 24.921 | 25.423 | 25.401 | 24.206 | 25.318 | 22.439 | 24.055 | 25.297 | 24.194 | 22.920 | 26.001 | 24.594 | **26.174** | 25.405 |
| FID↓ | 93.564 | 22.508 | 20.676 | 17.080 | 22.790 | 18.295 | 19.654 | 19.288 | 32.628 | 22.547 | 14.248 | 21.393 | 25.6026 | **13.910** |
| NIQE↓ | 9.1407 | 6.7775 | 6.7324 | 5.3440 | 6.7038 | 5.3992 | 6.1638 | 6.4053 | 8.1487 | 5.4556 | 5.6228 | 6.3571 | 6.5709 | **5.3300** |
| LPIPS↓ | 0.5953 | 0.2956 | 0.2823 | 0.2702 | 0.2965 | 0.2695 | 0.3052 | 0.2689 | 0.3447 | 0.3011 | 0.2413 | 0.2623 | 0.2851 | **0.2395** |

natural and accurate restoration in both overall color retention and local texture generation, attributed to the effective use of high-performance generative models. Quantitative comparisons are presented in Table 1. Our method demonstrates competitive performance on perceptual metrics such as FID, NIQE, and LPIPS. Throughout the restoration process, our approach achieves results that are more natural and consistent with facial priors.

**Evaluation on Real-world Datasets.** As illustrated in Figure 5, our method demonstrates exceptional robustness in handling real-world degradations, consistently producing HQ face images. In contrast, diffusion-based methods like FlowIE and PMRF often result in unnatural facial structures. While codebook-based approaches can sometimes generate natural-looking faces, they frequently introduce substantial color and texture artifacts. Our method, however, excels in predicting local textures with fewer imperfections compared to other diffusion-based techniques. This advantage is attributed to our pseudo-hashing module, which enables the modeling of a more continuous optimal learning path. Table 2 presents a detailed comparison of our approach against existing SOTA methods, highlighting our superior performance across perceptual quantitative metrics. Particularly

on the challenging Wider dataset, which features severe and complex degradations, our method achieves leading results on no-reference metrics such as FID and NIQE. These findings underscore the robustness and generalizability of our approach in a wide range of real-world scenarios.

### 5.2. Ablation Studies

**Different between PHM and data augmentation.** Data augmentation involves using certain properties of images, such as rotation and brightness, to augment the actual numerical distribution of training data without altering the semantic information of the images. In contrast, the strategy design in pseudo-hashing module (PHM) is intended to satisfy the boundary constraints in SB models, requiring the direct or indirect involvement of degradation representations, and this hashing design is reversible.

**Guidance Module.** In this study, we systematically validate the efficacy of the guidance module, encompassing both conditional and degradation-aware components, through a series of ablation experiments. As delineated in Table 3, experiments (a)-(c) rigorously analyze the impact of incorporating these guidance mechanisms on restoration perfor-

*Table 2.* **Quantitative comparison on real-world datasets**. **Bold** indicates the best performance. The non-reference metric NIQE assesses the quality of restoration, while the distribution gap between FFHQ and results is quantified by FID.

| Dataset | Type-based | LFW | | CelebChild | | WebPhoto-Test | | Wider | |
| Method | | NIQE↓ | FID↓ | NIQE↓ | FID↓ | NIQE↓ | FID↓ | NIQE↓ | FID↓ |
|---|---|---|---|---|---|---|---|---|---|
| LQ | - | 6.4582 | 140.64 | 6.6542 | 140.46 | 10.0806 | 174.52 | 12.0099 | 174.52 |
| GPEN | GAN | 6.0327 | 54.51 | 6.6999 | 118.66 | 6.6234 | 97.42 | 6.2069 | 97.42 |
| GFPGAN | GAN | 6.2330 | 53.53 | 7.0313 | 118.76 | 7.1427 | 104.63 | 6.9093 | 104.63 |
| VQFR | Codebook | 5.5092 | 54.08 | 6.5187 | 116.65 | 6.1202 | 88.52 | 5.4927 | 88.52 |
| CodeFormer | Codebook | 5.3945 | 54.82 | 6.4258 | 114.67 | 5.8507 | 87.49 | 5.2382 | 87.49 |
| DifFace | Diffusion | 6.8339 | 66.89 | 6.7321 | 119.07 | 7.3431 | 97.60 | 7.9364 | 97.60 |
| DiffBIR | Diffusion | 5.8180 | 44.42 | 6.2220 | 110.48 | 6.3345 | 91.95 | 5.6994 | 91.95 |
| DR2 | Diffusion | 6.2578 | 54.29 | 7.0488 | 123.53 | 8.2475 | 116.40 | 8.6105 | 116.40 |
| PGDiff | Diffusion | **5.1049** | 47.19 | 5.8718 | 111.72 | 5.3651 | 89.69 | 4.9502 | 89.69 |
| PMRF | Flow | **5.1464** | 54.97 | **5.5044** | 108.30 | 5.4790 | **83.76** | 5.2207 | 83.76 |
| FlowIE | Flow | 5.7388 | **51.03** | 6.3734 | 112.69 | 6.5751 | 92.94 | 5.5997 | 92.94 |
| Vanilla-I2SB | Bridge | 5.8074 | 64.43 | 7.1276 | 123.82 | 6.9031 | 108.03 | 6.0377 | 108.03 |
| P-I2SB | Bridge | **5.2125** | **54.39** | **5.6425** | **108.07** | **5.2329** | **84.68** | **4.9056** | **69.62** |

*Table 3.* **Ablation studies** on CelebA-Test. (a) donates Vanilla-I2SB as the baseline, (b)-(c) compare different condition guidance, and (d)-(f) compare different Pseudo-Hashing strategies. "lq" donates LQ images as the condition in forward and reverse process and "da" donates degradation-aware representation as condition.

| Method | Condition | | Hashing | Metrics | | |
| | lq | da | strategy | NIQE↓ | FID↓ | LPIPS↓ |
|---|---|---|---|---|---|---|
| (a) | | | | 25.6026 | 6.5709 | 0.2851 |
| (b) | ✓ | | | 25.9522 | 6.4849 | 0.2904 |
| (c) | ✓ | ✓ | | 25.4932 | 6.3898 | 0.2945 |
| (d) **ours** | ✓ | ✓ | Noise | 18.2646 | 5.7595 | 0.2678 |
| (e) **ours** | ✓ | ✓ | Cat | 13.9109 | 5.3300 | 0.2395 |
| (f) **ours** | ✓ | ✓ | Res | 14.2941 | 5.4401 | 0.2431 |

mance. Leveraging three established perceptual metrics, including FID, NIQE, and LPIPS, we quantify the contributions of each guidance component. The model that integrates both types of guidance achieves the best restoration results, indicating that learning the process between two distributions directly on BFR using SB is not optimal. Appropriate guidance conditions can enhance the ability of the model to fit the inverse process.

**Pseudo-Hashing Strategy.** Finally, and most importantly, we demonstrate the significant improvements brought by the pseudo-hashing strategy in the comparisons between experiments (c) and (d)-(f). There are marked enhancements in NIQE, FID and LPIPS. The visual results in Appendix G also reveal that the degradation artifacts are noticeably reduced in (a)-(c), with our P-I2SB exhibiting superior content preservation and local texture generation.

## 6. Discussion and Future Work

**Advantages.** This work investigates why image-to-image transfer methods like I2SB fail in blind inverse problems. It identifies the core issue as the limitations imposed by the optimality and reversibility of SB solutions, which result in poor restoration. Inspired by spatial geometry, we propose a straightforward correction strategy for training image pairs,

termed P-I2SB. This strategy circumvents the conflict between blind inverse problems and the original definition of SB, which often leads to a lack of feasible solutions. By deeply exploring and leveraging the capability of bridge-based models to learn optimal transport, P-I2SB improves restoration performance. Extensive experiments on the practical BFR demonstrate that P-I2SB significantly improves the restoration capabilities of the Schrödinger Bridge.

**Limitations and Future Work.** Our approach is limited by the inability of artificially generated degradation processes to fully replicate real-world scenarios, such as watermarks, where restoration remains suboptimal. To address this, we propose three strategies: using segmentation to isolate primary subjects and reduce interference, expanding degradation types in training datasets to enhance restoration through supervised learning, and employing contrastive learning to improve degradation-aware information extraction from real-world datasets via unsupervised learning. Additionally, the dominance of frontal face images in the training data limits the model's ability to restore faces in extreme poses. Future work will integrate methods like 3D Morphable Models (3DMMs) to extract pose information, aiding the learning from less represented samples and improving robustness across diverse facial orientations.

## 7. Conclusion

In this paper, we introduced P-I2SB for the challenge of blind face restoration, structured across two stages: the Pseudo-Hashing Module (PHM) and the Schrödinger Bridge Module (SBM). Our theoretical analysis identified fundamental limitations of traditional Schrödinger Bridge (SB) solutions in handling blind problems and led to the development of three feasible pseudo-hashing strategies. Experimental results demonstrate the superior restoration performance on both synthetic and real-world datasets. This work offers a novel perspective on resolving challenges in blind problems by effectively computing the relationship between two distributions.

## Acknowledgements

This paper is supported by National Key R&D (Research and Development) Program of China (2021YFA1000403), the National Natural Science Foundation of China (Nos. 12431012, U23B2012, 12471308) and Beijing Natural Science Foundation (1254050).

## Impact Statement

This project aims to provide an effective generative approach to address the problem of blind face restoration (BFR). Our method theoretically investigates the ill-posedness of Schrödinger Bridge solutions in the context of BFR, proposing a novel framework that directly uses degraded images as conditions for restoration through preprocessing of data distributions, eliminating the need for any feature extraction design. However, like other generative technologies, there is potential for misuse. Malicious entities could combine this method with target generative models to produce misleading or harmful 2D facial images. Addressing these ethical issues is crucial, and future research in 2D generation must prioritize understanding and mitigating these risks to ensure the responsible application of these technologies.

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

## A. Overview

In this Appendix, additional proof, experimental details and experimental results are provided, including:

- Details about the connection between different versions of OMT problem (in Sec. B);

- Details about the proof of theorem 4.1 (in Sec. C);

- Details about experiments setting (in Sec. D);

- Details about preprocessing data (in Sec. E);

- Details about loss analysis (in Sec. F);

- More ablation results (in Sec. G);

- Qualitative comparisons with SOTA methods in CelebA-Test and real-world datasets (in Sec. H and Sec. I);

- Qualitative results about failure samples (in Sec. J).

## B. OMT

In the 1940s, Kantorovich studied a "relaxed" version of the Monge's problem (Chen et al., 2021b):

$$\inf_{\pi \in \Pi(\nu_0, \nu_1)} \int_{\mathbb{R}^{2d}} c(x, y)\pi(\mathrm{d}x\mathrm{d}y). \tag{20}$$

Here, $\Pi(\nu_0, \nu_1)$ is the probability distributions on $\mathbb{R}^{2d}$ with marginals $\nu_0$ and $\nu_1$. Indeed, $\Pi(\nu_0, \nu_1)$ always contain the product measure $\nu_0 \otimes \nu_1$.

Though there are some different vrsions of OMT problem, they have some connection. If $c(x, y) = ||x - y||^2$ and if $\nu_1$ does not give mass to sets of dimension $\leq n_1$, there exists a unique optimal $\pi^*$ (Kantorovich) induced by a unique (Monge) map $T^*$, such that $\pi^* = (I \times T^*)\#\nu_0$ where $I$ is the identity map (Villani, 2021).

Let

$$\mathcal{W}_2^2 := \inf_{\pi \in \Pi(\nu_0, \nu_1)} \int_{\mathbb{R}^{2d}} ||x - y||^2 \pi(\mathrm{d}x\mathrm{d}y) \tag{21}$$

called as Kantorovich–Wasserstein quadratic distance. $\mathcal{W}_2^2$ is also the minimum value of the following optimal problem (Benamou & Brenier, 2000) written as problem (2). Let $\{\mu_t^*; 0 \leq t \leq 1\}$ and $\{v^*(t, x); (t, x) \in [0, 1] \times \mathbb{R}^d\}$ be optimal for (2). And now, $\mu_t^* = [(1 - t)I + tT^*]\#\nu_0$.

## C. Proof

**The proof of theorem 4.1.** We proof the theorem for a common Markov path measure $P \in \mathbb{D}$ whose coordinate process has the SDE representation (9). The evolution its density function $p_t(x)$ can be characterized by the Fokker Plank equation on time interval $(0, 1)$:

$$\frac{\partial p_t(x)}{\partial t} = -\nabla \cdot (\alpha_t(x)p_t(x)) + \frac{1}{2}\beta_t \Delta p_t(x). \tag{22}$$

By disintegration formula (10), we have

$$P_t(z) = \int_{\mathbb{R}^{2d}} P_t^{xy}(z) P_{01}(\mathrm{d}x\mathrm{d}y). \tag{23}$$

We can get the disintegration formula of the density function:

$$p_t(z) = \int_{\mathbb{R}^{2d}} p_t^{xy}(z) P_{01}(\mathrm{d}x\mathrm{d}y). \tag{24}$$

Suppose all the density functions are good enough, we can do some differential operations on equation (24):

$$(a): \quad \frac{\partial p_t(z)}{\partial t} = \frac{\partial \int_{\mathbb{R}^{2d}} p_t^{xy}(z) P_{01}(\mathrm{d}x\mathrm{d}y)}{\partial t}$$
$$= \int_{\mathbb{R}^{2d}} \frac{\partial p_t^{xy}(z)}{\partial t} P_{01}(\mathrm{d}x\mathrm{d}y), \tag{25}$$

$$(b): \quad \nabla \cdot (\alpha_t(z)p_t(z)) = \nabla \cdot \int_{\mathbb{R}^{2d}} \alpha_t(z) p_t^{xy}(z) P_{01}(\mathrm{d}x\mathrm{d}y)$$
$$= \int_{\mathbb{R}^{2d}} \nabla \cdot (\alpha_t(z)p_t^{xy}(z)) P_{01}(\mathrm{d}x\mathrm{d}y), \tag{26}$$

$$(c): \quad \Delta p_t(z) = \nabla \cdot \nabla p_t(z)$$
$$= \nabla \cdot \int_{\mathbb{R}^{2d}} \nabla p_t^{xy}(z) P_{01}(\mathrm{d}x\mathrm{d}y)$$
$$= \int_{\mathbb{R}^{2d}} \Delta p_t^{xy}(z) P_{01}(\mathrm{d}x\mathrm{d}y). \tag{27}$$

Then, let $(a) + (b) - \frac{1}{2}(c)$:

$$\frac{\partial p_t(z)}{\partial t} + \nabla \cdot (\alpha_t(z)p_t(z)) - \frac{1}{2}\Delta p_t(z) = 0$$
$$\Leftrightarrow \int_{\mathbb{R}^{2d}} \frac{\partial p_t^{xy}(z)}{\partial t} + \nabla \cdot (\alpha_t(z)p_t^{xy}(z)) - \frac{1}{2}\Delta p_t^{xy}(z) P_{01}(\mathrm{d}x\mathrm{d}y) = 0. \tag{28}$$

Until now, we stress that what we consider is a boundary value problem. That is, we want to find some boundary conditions subject to our requirement which is to find a suitable $P_{01}$ so that the following equation can hold:

$$\frac{\partial p_t^{xy}(z)}{\partial t} = -\nabla \cdot (\alpha_t(z)p_t^{xy}(z)) + \frac{1}{2}\Delta p_t^{xy}(z). \tag{29}$$

One obvious choice is the Dirac distribution. Let $p_{01}(x,y)\mathrm{d}x\mathrm{d}y := P_{01}(\mathrm{d}x\mathrm{d}y)$. In the sense of Dirac distributions where $p_0(x) = \delta_{x_0}(x) := \delta(x - x_0)$ and $p_1(y) = \delta_{y_1}(y) := \delta(y - y_1)$, we can get:

$$\delta(x - x_0)\delta(y - y_1) = p_{01}(x,y)$$
$$= p_0(x)p(y|x) = p_0(x)p_1(y) = \delta_{x_0}(x)\delta_{y_1}(y). \tag{30}$$

We concluded that the bridges of a path between pairs $(x, y)$ can induce an identical SDE over time interval $(0, 1)$ when we take the boundary distribution as Dirac distributions centered at $x$ and $y$.

## D. Implementation Details

We implement P-I2SB using PyTorch, leveraging eight 32GB Tesla V100 GPUs. The training process employs the Unet network as the predictor. The Adam optimizer is utilized with a learning rate of $1 \times 10^{-4}$. The training process comprises 40000 iterations with a batch size of 64.

## E. Pre-processing data

Since our P-I2SB relies on supervised training, it necessitates paired low-quality (LQ) and high-quality (HQ) images. To simulate LQ images representative of real-world scenarios, we employ a degradation model generated through random sampling. Its generation formula (Zhang et al., 2018a; Wang et al., 2021) is presented in Eq.(31), where $x$ is the HQ image, $k_\sigma$ is the Gaussian blur kernel, $r$ represents the down-sampling scale factor, and $q$ represents the JPEG compression of

*Table 4.* Degradation parameters in training stage.

| Method | Blur list | Blur size | Blur sigma | Kernel prob | Noise range | Down sampling | Jpeg quality | Gray prob |
|---|---|---|---|---|---|---|---|---|
| GPEN | 'iso', 'aniso' | 41 | [0.1, 10] | [0.5, 0.5] | [0, 20] | [0.8, 8] | [60, 100] | 0.2 |
| GFPGAN | 'iso', 'aniso' | 41 | [0.1, 10] | [0.5, 0.5] | [0, 20] | [0.8, 8] | [60, 100] | 0.01 |
| P-I2SB | 'iso', 'aniso' | 41 | [0.1, 10] | [0.5, 0.5] | [0, 20] | [0.8, 8] | [60, 100] | 0.2 |

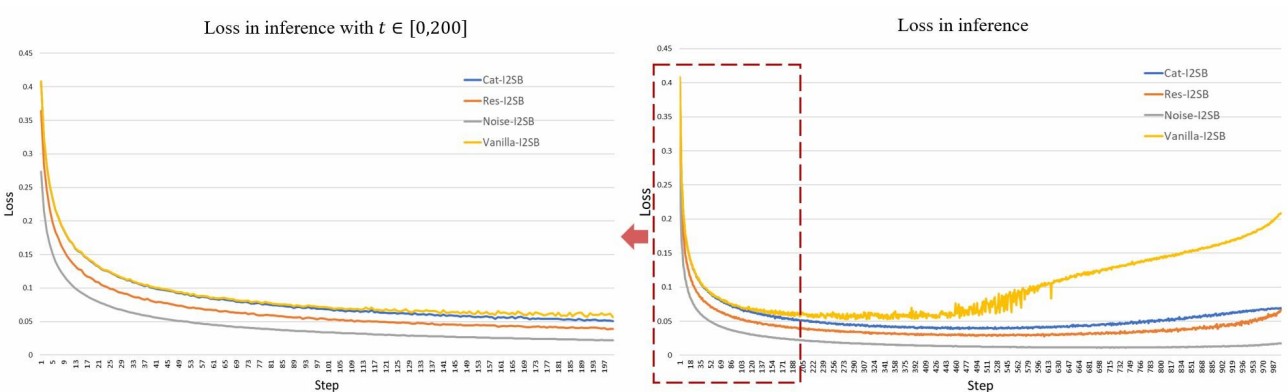

*Figure 6.* **Loss in Inference**. The right shows the loss across different steps during inference for 64 batch images, from $step = T$ to $step = 0$. The left resents a zoomed-in view of the curve from $step = 200$ to $step = 0$.

the image with quality factor $q$. To ensure direct comparability with the experimental results of recent BFR methods, we randomly sample the parameters $\sigma$, $r$, $\delta$, $q$ from {0.1: 10}, {0.8: 8}, {0: 20}, {60: 100}, respectively.

$$y = [(x \otimes k_\sigma) \downarrow_r + n_\delta]_{\text{JPEG}_q} \tag{31}$$

This parameter selection is consistent with the established experimental settings in the field, as shown in Table 4. To enhance color adaptation and mitigate overfitting, we introduce a gray color probability mechanism and apply horizontal flip transformations to the training data. The specific degradation handling functions are included in the python file.

## F. Numerical Experiments

Figure 6 illustrates the variation in loss at different steps during the reverse inference process. It can be observed that the loss for Vanilla-I2SB exhibits significant fluctuations, with substantial loss occurring at the initial stages of inference. This indicates that, under complex degradation conditions, the Schrödinger Bridge model fails to learn a feasible solution, leading to severe oscillations at the endpoints of the constraints. In contrast, the three loss curves following the pseudo-hashing strategy preprocessing show a noticeably smoother loss at the early stages of inference.

## G. More Ablation Results

Figure 7 illustrates the results of the ablation study through visualizations. Panel (a) depicts the Vanilla-I2SB method, serving as the baseline for comparison. Panel (b) utilizes low-quality (LQ) images as conditional guidance, while panel (c) incorporates both LQ images and degradation representation for enhanced guidance. Panels (d) through (f) demonstrate the application of three distinct pseudo-random strategies. The visualizations indicate that although panels (a) to (c) effectively restore the general outlines of the images, they fail to capture fine local texture details, leading to an overall smooth and averaged appearance. This inadequacy is particularly evident in the restoration of critical facial features, such as the eyes, where accuracy is compromised.

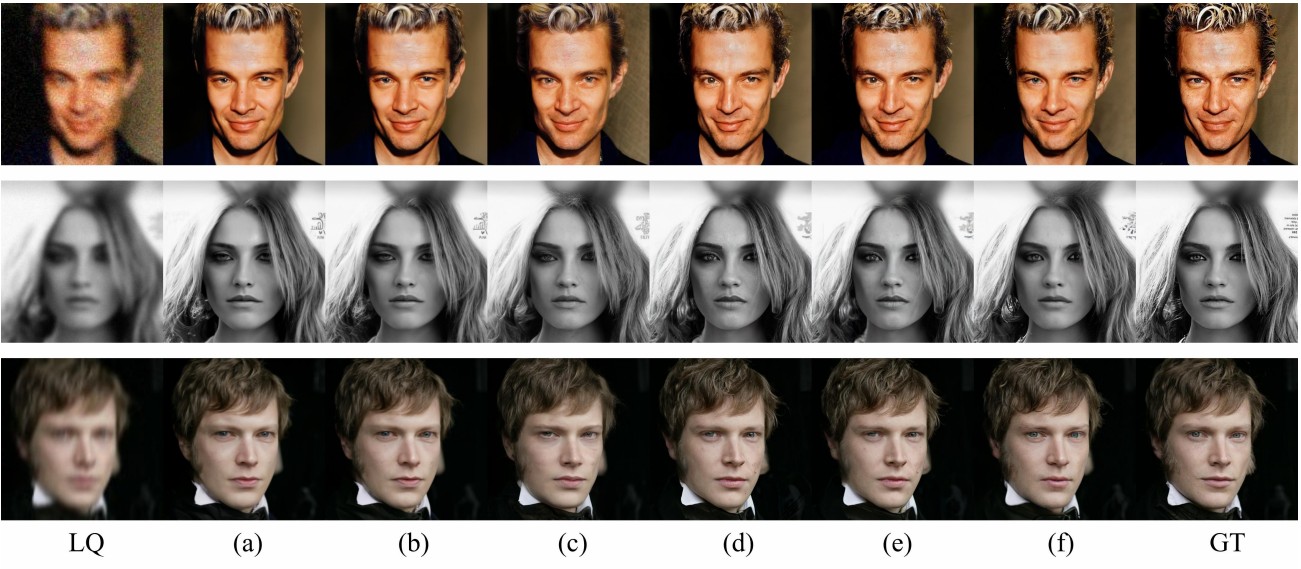

LQ    (a)    (b)    (c)    (d)    (e)    (f)    GT

*Figure 7.* **Ablation studies on CelebA-Test**. (a) donates Vanilla-I2SB as baseline, (b)-(c) compare different condition guidance, and (d)-(f) compare different Pseudo-Hashing strategies. **Zoom in for best view**.

## H. Qualitative Comparisons in CelebA-Test

In Figure 8, we present a comprehensive comparison of P-I2SB with several state-of-the-art methods, including GFP-GAN (Wang et al., 2021), GPEN (Yang et al., 2021), VQFR (Gu et al., 2022), CodeFormer (Zhou et al., 2022), Restore-Former (Wang et al., 2022), DMDNet (Li et al., 2022), DAEFR (Tsai et al., 2023), DifFace (Yue & Loy, 2022), DiffBIR (Lin et al., 2023), DR2 (Wang et al., 2023), PGDiff (Yang et al., 2024), FlowIE (Zhu et al., 2024) and PMRF (Ohayon et al., 2024). Our results indicate that P-I2SB excels in preserving the original identity and facial expressions of the images while generating natural and plausible texture details. This method surpasses other approaches in terms of visual fidelity and detail accuracy, offering superior performance in maintaining both the overall appearance and the intricate nuances of facial features.

## I. Qualitative Comparisons in Real-world

Figures 9, 10, 11, and 12 present a comparative analysis of P-I2SB across four real-world datasets: CelebChild, LFW, WedPhoto-Test, and Wider. Figure 13 shows comparative results of low-quality images with serious degradation in real-world datasets. These comparisons are made against several state-of-the-art methods, including GPEN (Yang et al., 2021), GFPGAN (Wang et al., 2021), VQFR (Gu et al., 2022), CodeFormer (Zhou et al., 2022), RestoreFormer (Wang et al., 2022), DMDNet (Li et al., 2022), DAEFR (Tsai et al., 2023), DifFace (Yue & Loy, 2022), DiffBIR (Lin et al., 2023), DR2 (Wang et al., 2023), PGDiff (Yang et al., 2024), FlowIE (Zhu et al., 2024), PMRF (Ohayon et al., 2024) and Vanilla-I2SB (Liu et al., 2023). The results demonstrate that P-I2SB generates more coherent and detailed outputs by effectively utilizing the Pseudo-Hashing Module (PHM) and Schrödinger Bridge Module (SBM). This approach circumvents the indirect process of extracting features from LQ images by directly using the raw LQ images in place of Gaussian noise as one endpoint of the data distribution in the generative diffusion model. This ensures the existence of a theoretically optimal solution. The pseudo-hashing strategy significantly improves the feasibility of solutions in blind inverse problems based on the Schrödinger Bridge method. The generated images excel in preserving facial structure and generating local texture details.

## J. Failure Samples

Figure 14 illustrates the failure cases of P-I2SB. Specifically, when real-world images contain watermarks, the inference results of P-I2SB still exhibit residual artifacts. This issue primarily arises because the model is trained on synthetically generated degradation datasets, which fail to fully capture the complexity of real-world degradations. To address this

limitation, our future research will focus on enhancing the robustness of the Schrödinger Bridge method by utilizing unsupervised learning to augment the pseudo-hashing strategy, thereby extending its applicability to a broader range of blind inverse problems.

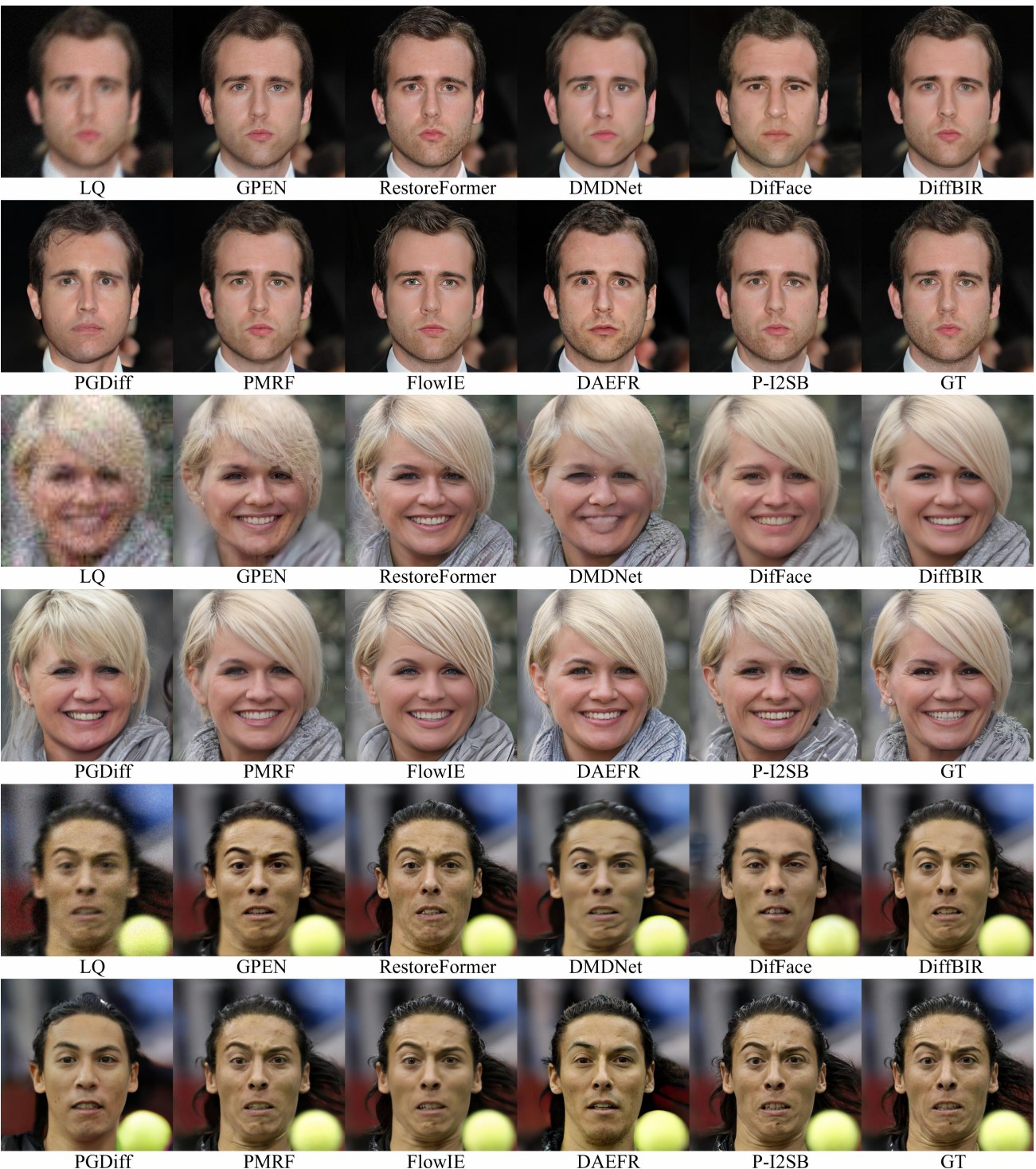

*Figure 8.* **Qualitative comparisons on CelebA-Test**. Our P-I2SB demonstrates superior performance in both detail enhancement and hue preservation, particularly on inputs with severe degradation. **Zoom in for best view**.

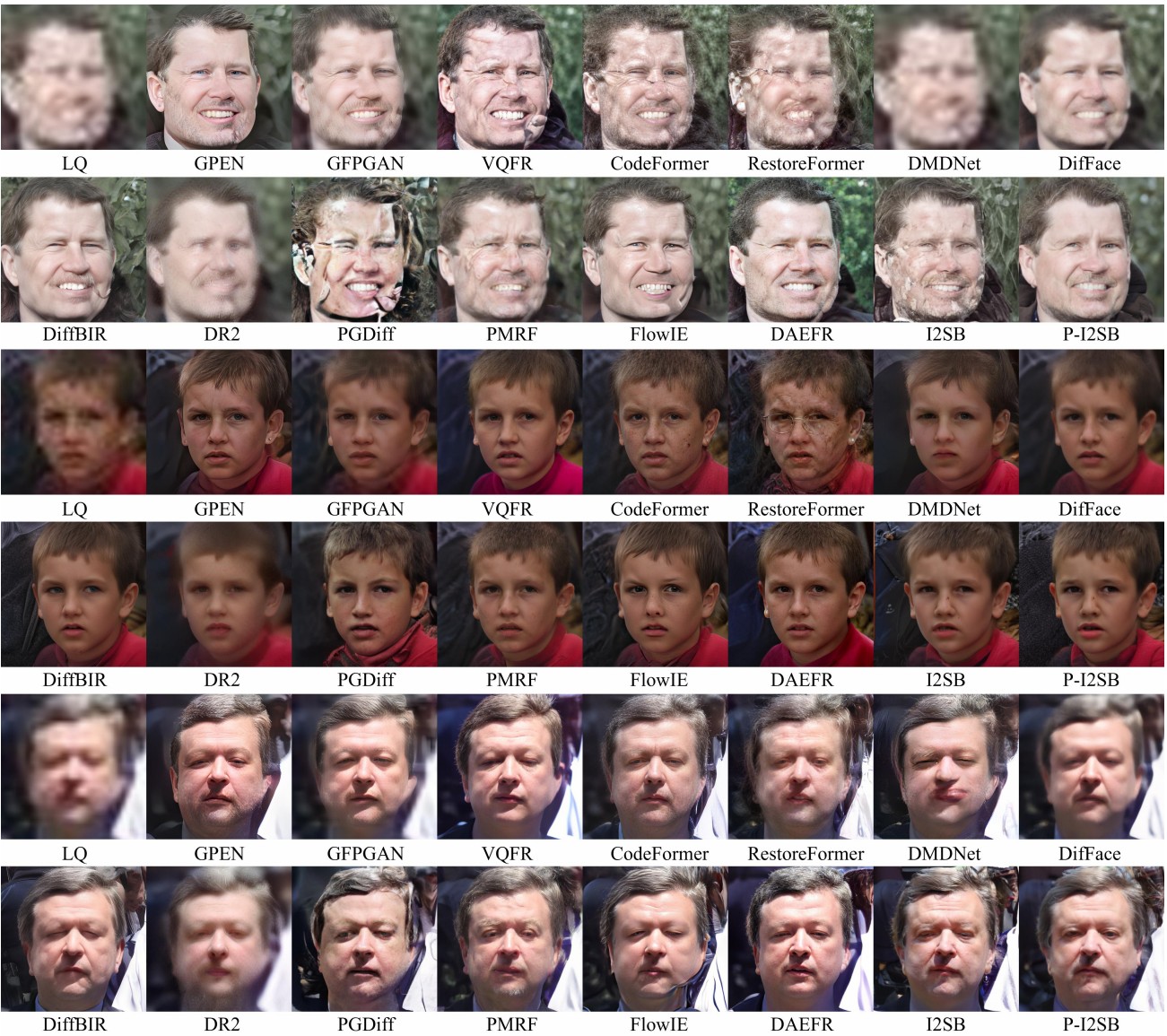

*Figure 9.* **Qualitative comparisons on Wider**. In terms of both detail enhancement and hue preservation, our P-I2SB outperforms, particularly on inputs suffering from severe degradation. **Zoom in for best view**.

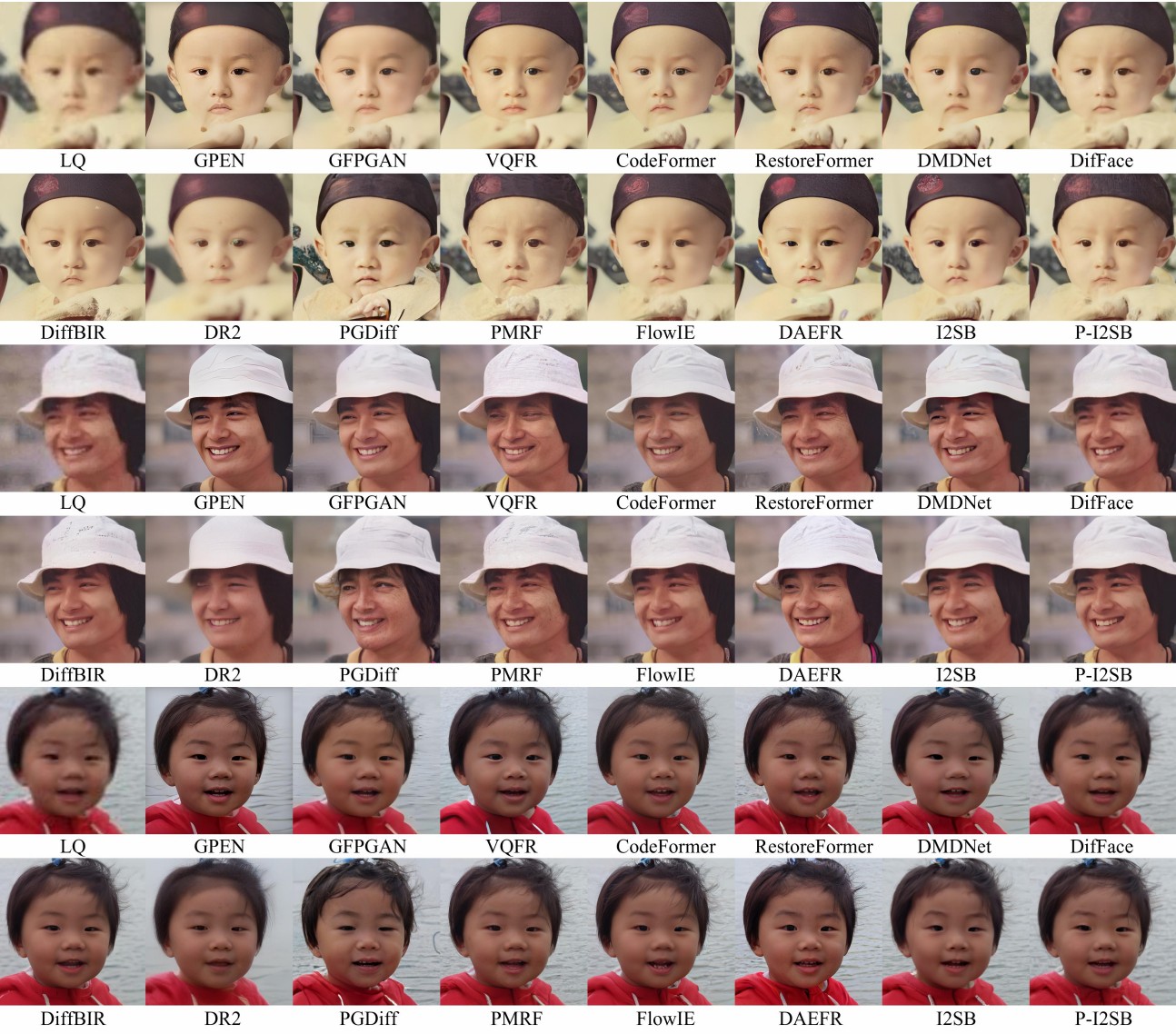

*Figure 10.* **Qualitative comparisons on WebPhoto**. Demonstrating superior capability, our P-I2SB significantly improves detail and hue retention, even on inputs with extreme degradation. **Zoom in for best view**.

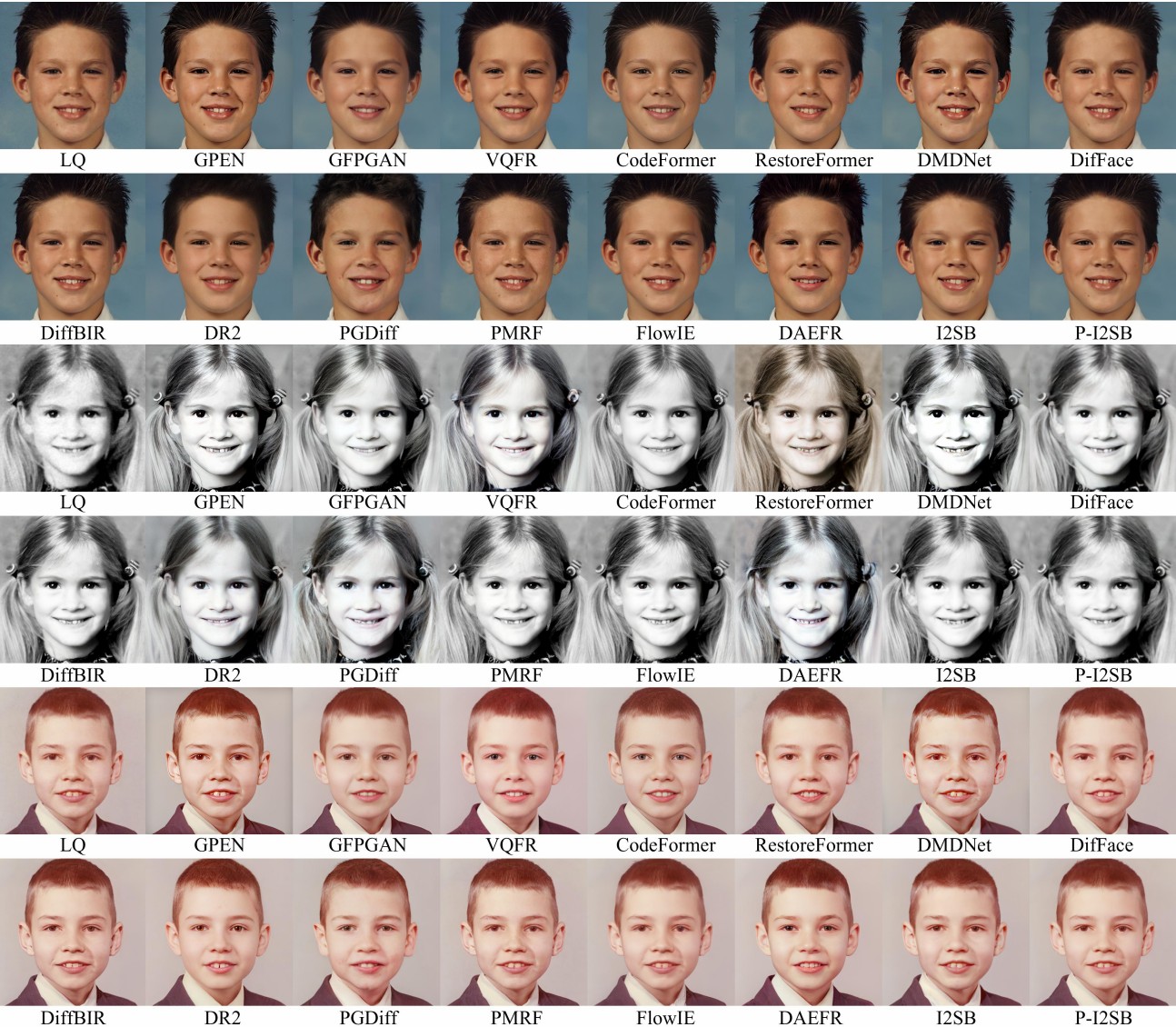

*Figure 11.* **Qualitative comparisons on CelebChild-Test**. Our P-I2SB excels in both enhancing fine details and preserving hues, especially when dealing with severely degraded inputs. **Zoom in for best view**.

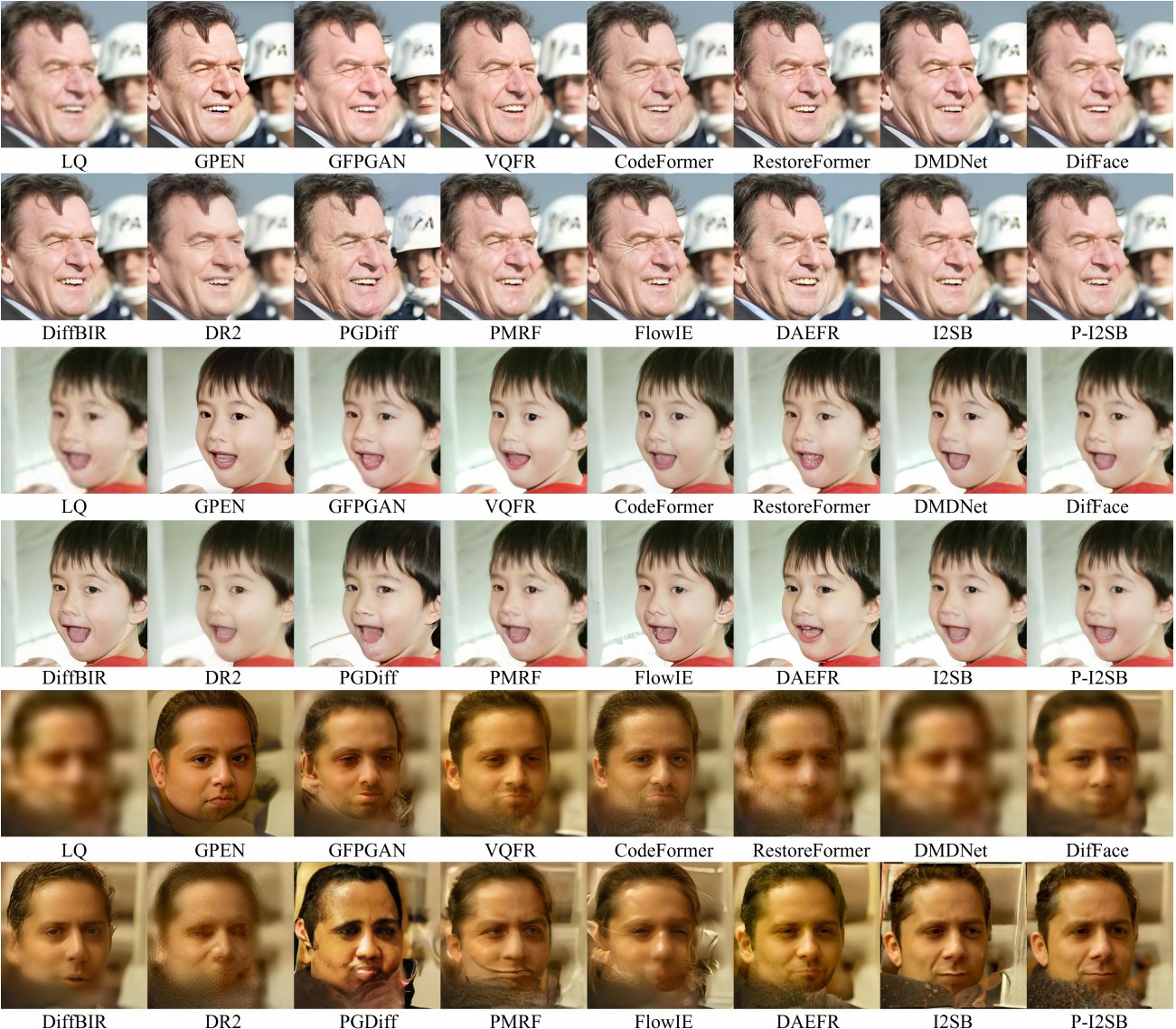

*Figure 12.* **Qualitative comparisons on LFW**. The P-I2SB showcases exceptional performance in detail refinement and color fidelity, particularly on heavily deteriorated inputs. **Zoom in for best view**.

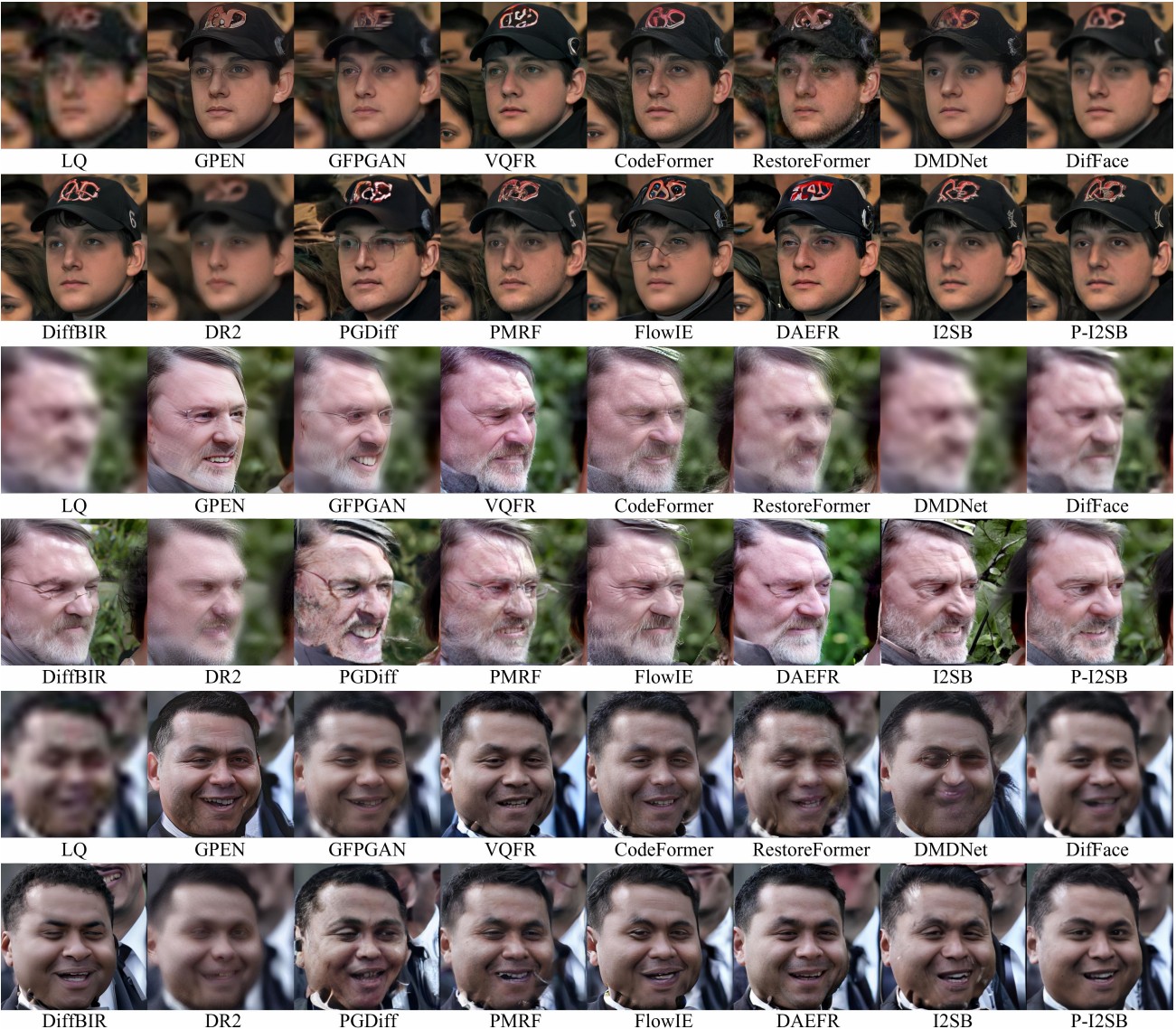

*Figure 13.* **Qualitative comparisons on serious degradation in real-world dataset**. The P-I2SB showcases exceptional performance in detail refinement and color fidelity, particularly on heavily deteriorated inputs. **Zoom in for best view**.

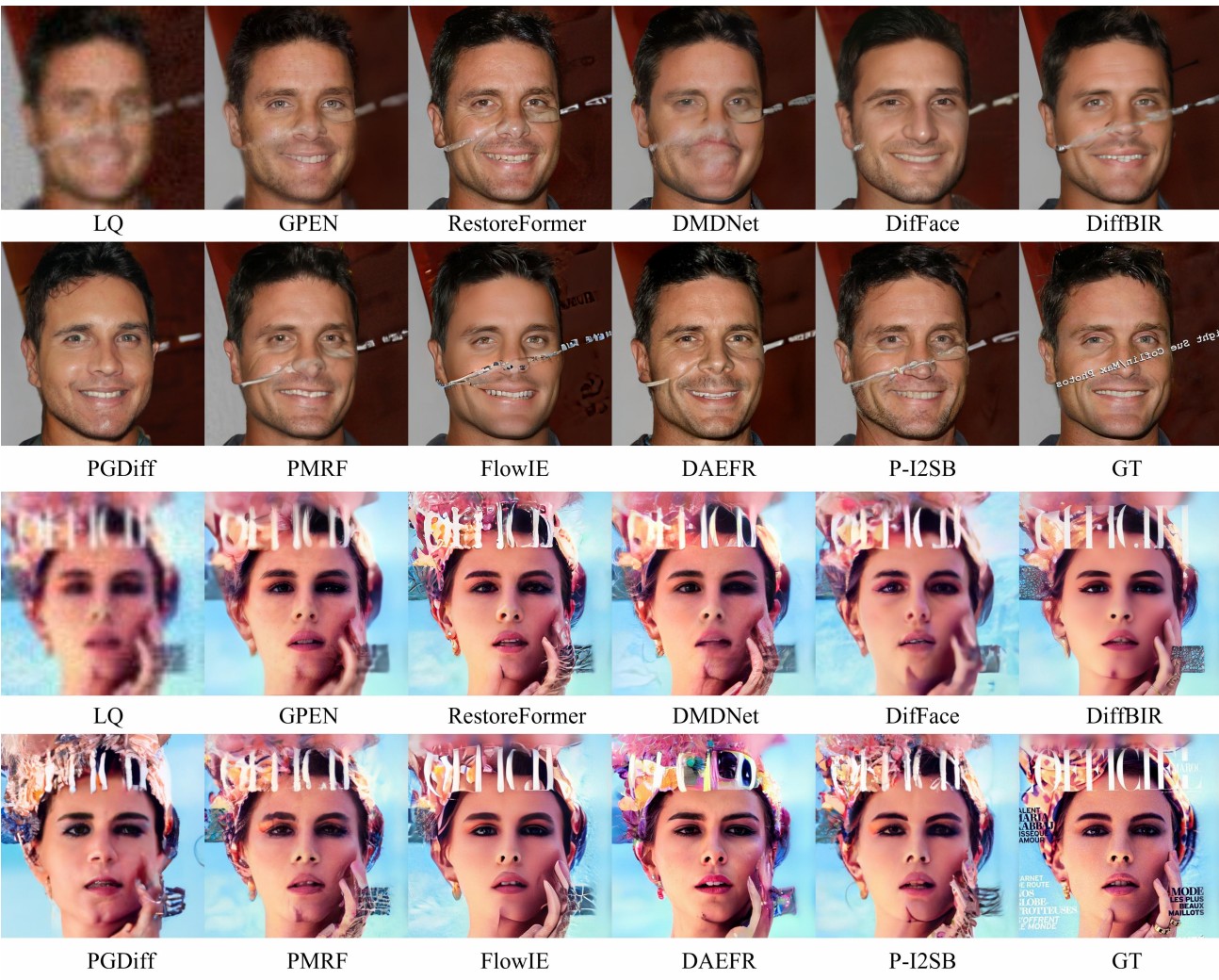

Figure 14. **Failure samples in CelebA-Test**. Degradation artifacts such as watermarks can seriously affect the restoration results. **Zoom in for best view**.

