# OpenReview forum: "Feature out! Let Raw Image as Your Condition for Blind Face Restoration"
_ICML.cc/2025/Conference — ICML 2025 poster_

### Official Review · Reviewer_UKRe · 2025-03-11

**Overall Recommendation:** 3

**Summary:**

This paper proposes the Pseudo-Hashing Image-to-image Schrodinger Bridge (P-I2SB) framework to enhance the restoration potential of Schrodinger Bridge (SB) by correcting data distributions and effectively learn the optimal transport path between any two data distributions. This approach preprocesses HQ images during training by hashing them into pseudo-samples according to a rule related to LQ images. This guarantees optimal and reversible solutions in SB, enabling the inference process to learn effectively and allowing P-I2SB to achieve state-of-the-art results in BFR.

## update after rebuttal: The rebuttal has addressed certain formatting issues in the manuscript. I respectfully maintain my original rating.

**Claims And Evidence:**

Yes.

**Essential References Not Discussed:**

None.

**Experimental Designs Or Analyses:**

The authors conducted extensive experiments to validate their proposed method.

**Methods And Evaluation Criteria:**

It is the first work to introduce the Pseudo-Hashing Module (PHM) and the Schrodinger Bridge Module (SBM) theories into blind face restoration.
Experimental results demonstrate the superior restoration performance on both synthetic and real-world datasets.

**Other Comments Or Suggestions:**

In Table 1, the reference numbers for SoTA methods should be listed.

**Other Strengths And Weaknesses:**

None.

**Questions For Authors:**

None.

**Relation To Broader Scientific Literature:**

This paper complements the exploration of blind face restoration, making a certain contribution.

**Theoretical Claims:**

Yes.

---

> ### Author Rebuttal · Authors · 2025-04-01
>
> Thank you for your careful review of the paper structure and formatting, which is greatly appreciated.
> > Q1. In Table 1, the reference numbers for SoTA methods should be listed.
>
> - Due to the ICML citation format not using numbers, the references were too lengthy to include directly in the table. Instead, we noted the references in the Comparison Methods section of Sec. 5.1. We have now supplemented the references and adjusted the font size in the table to ensure that they can be properly displayed in the main paper in future versions.
> > Sec.5.1 (Line 361 - Line 370) **Comparison Methods** We compare Pseudo-Hashing with recent BFR methods, including PSFRGAN (Chen et al., 2021a), GFPGAN (Wang et al., 2021), GPEN (Yang et al., 2021), VQFR (Gu et al., 2022), CodeFormer (Zhou et al., 2022), RestoreFormer (Wang et al., 2022), DMDNet (Li et al., 2022), DAEFR (Tsai et al., 2023), DifFace (Yue & Loy, 2022), DR2 (Wang et al., 2023), PGDiff (Yang et al., 2024), DiffBIR (Lin et al., 2023), PMRF (Ohayon et al., 2024), FlowIE (Zhu et al., 2024) and I2SB (Liu et al., 2023) .
>
> | Metrics        | Input  | GPEN | GFP | Restore | DMDNet | DAEFR | DifFace | DiffBIR | DR2 | PGDiff | PMRF | FlowIE | I2SB | **P-I2SB** |
> |----------------|--------|------|-----|---------|--------|-------|---------|---------|-----|--------|------|--------|------|------------|
> |                |        | *CVPR* | *CVPR* | *CVPR* | *TPAMI* | *ICLR* | *TPAMI* | *ECCV* | *CVPR* | *NIPS* | *ICML* | *CVPR* | *ICML* |            |
> |                |        | (Yang et al., 2021) | (Wang et al., 2021) | (Wang et al., 2022) | (Li et al., 2022) | (Tsai et al., 2023) | (Yue & Loy, 2022) | (Lin et al., 2023) | (Wang et al., 2023) | (Yang et al., 2024) | (Ohayon et al., 2024) | (Zhu et al., 2024) | (Liu et al., 2023) |            |
> | SSIM ↑         | 0.6460 | 0.6777 | _0.6827_ | 0.6219  | 0.6727 | 0.5892  | 0.6494  | 0.6570  | 0.6554 | 0.6220  | 0.6815 | 0.6479 | **0.7047** | 0.6581     |
> | PSNR ↑         | 24.921 | 25.423 | 25.401   | 24.206  | 25.318 | 22.439  | 24.055  | 25.297  | 24.194 | 22.920  | _26.001_ | 24.594 | **26.174** | 25.405     |
> | FID ↓          | 93.564 | 22.508 | 20.676   | 17.080  | 22.790 | 18.295  | 19.654  | 19.288  | 32.628 | 22.547  | _14.248_ | 21.393 | 25.6026 | **13.910** |
> | NIQE ↓         | 9.1407 | 6.7775 | 6.7324   | _5.3440_ | 6.7038 | 5.3992  | 6.1638  | 6.4053  | 8.1487 | 5.4556  | 5.6228 | 6.3571 | 6.5709 | **5.3300** |
> | LPIPS ↓        | 0.5953 | 0.2956 | 0.2823   | 0.2702  | 0.2965 | 0.2695  | 0.3052  | 0.2689  | 0.3447 | 0.3011  | _0.2413_ | 0.2623 | 0.2851 | **0.2395** |      |

---

### Official Review · Reviewer_hmwC · 2025-03-12

**Overall Recommendation:** 3

**Summary:**

The authors present Pseudo-Hashing Image-to-Image Schrödinger Bridge (P-I2SB), a novel framework inspired by optimal mass transport. By correcting data distributions and effectively learning the optimal transport path between them, it enhances the restoration capabilities of Schrödinger Bridge (SB). Experimental results demonstrate that P-I2SB achieves state-of-the-art performance in blind face restoration (BFR), producing more natural textures compared to previous methods.

## Update after rebuttal

The authors have addressed most of my concerns. However, one point remains unclear: whether commonly used data augmentation techniques in image restoration—when applied alone, without the involvement of SwinIR or other external components—can help the Vanilla-I2SB model better identify the optimal path. Clarifying this would further illuminate the distinction between the effects of the PHM module and those of data augmentation. I will keep my score unchanged.

**Claims And Evidence:**

Yes

**Essential References Not Discussed:**

No.

**Experimental Designs Or Analyses:**

Yes, I have carefully reviewed the experimental design and the corresponding ablation studies, and I did not encounter any issues.

**Methods And Evaluation Criteria:**

Yes, the proposed method achieves state-of-the-art performance on the BFR task.

**Other Comments Or Suggestions:**

1. Lines 73 and 147 contain incorrect double quotation marks.
2. Line 164 references “problem (2),” but the text does not provide a corresponding explanation.

**Other Strengths And Weaknesses:**

Strengths:
1. The paper is well-structured and clearly presented.
2. The experiments are comprehensive, with thorough ablation studies.
3. The authors provide ample discussion, addressing both the method’s advantages and limitations.

Weaknesses:
1. While the authors claim that blind tasks do not align with the typical Monge optimal transport framework—due to the non-uniqueness of “optimal transport” between images—this purportedly contradicts the optimality principle of a reversible SB and complicates constructing a consistent I2SB model. However, their proposed approach appears more akin to a data augmentation technique. Similar to the data augmentation method in DiffBIR, pairing I2SB with such an approach could potentially achieve similar results, suggesting the method may be overly simplistic.
2. Although the authors assert that their approach learns more optimal transport paths than existing methods, they offer limited analyses or evidence to validate that the paths are indeed optimal. Additional experiments or analyses, such as those found in Appendix Figure 6, would help substantiate these claims.

**Questions For Authors:**

What is the significance of designing Cat-I2SB? After performing the cat operation, the resulting number of channels differs from other methods. If only the first three channels are retained, equation (12) can be simplified to $\(I_{hq}, I_{lq}^{d})$.

**Relation To Broader Scientific Literature:**

This paper builds upon the I2SB (ICML 2023) framework by integrating a novel pseudo-hashing preprocessing strategy.

**Theoretical Claims:**

Yes, The theoretical claims and proofs appear similar to those of I2SB. As I'm not an expert in this area, I did not notice any specific issues.

---

> ### Author Rebuttal · Authors · 2025-04-01
>
> Thank you for your comprehensive review and valuable insights.
> > Q1. Their approach resembles a data augmentation technique, like in DiffBIR, suggesting that pairing I2SB with it might yield similar results, indicating potential simplicity.
> - **Our P-I2SB is not a data augmentation method.** Data augmentation involves using certain properties of images, such as rotation and brightness, to augment the actual numerical distribution of training data without altering the semantic information of the images. In contrast, the strategy design in pseudo-hashing module (PHM) is intended to satisfy the boundary constraints in SB models, requiring the direct or indirect involvement of degradation representations, and this hashing design is reversible.
> - **DiffBIR** consists of two stages: first, the removal of degradation information, followed by generating a high-quality image that balances quality and identity. This is **completely different** from PHM. In our first stage (PHM), we do not remove any degradation information; instead, we rely on an SB-based model to learn indirect restoration relationships.
>
> > Q2. They provide limited evidence that the paths are optimal; additional analyses, like those in Appendix Figure 6, would help substantiate these claims.
> - **Existence of Optimal Path in P-I2SB**: We first establish the existence of the optimal path in our P-I2SB. Our model is constructed based on SB theory, as outlined in the Preliminaries (Sec.3). The consistency between Equation (7) and the optimal transport problem in Equation (2) reveals why the solution of the SB model is the optimal transport path.
> - **Monge's Problem**, linked to image restoration, seeks the optimal mapping $T$ between two distributions under a certain cost metric. While it can theoretically map low-quality to high-quality images, its practical scope is limited by mapping constraints. Therefore, the optimal transport problem introduces a version that finds the optimal joint distribution, ensuring a solution always exists. This is depicted in Equation (2). In BFR problems, the relationship between low- and high-quality images is more complex. To leverage the existing I2SB model, as discussed in Preliminaries (Sec.3), we propose PHM to address these complexities.
> - **Additional Experiments and Analyses**: To analyze the superiority of the path found by our P-I2SB, we compared P-I2SB with the baseline method on the loss analyses along the path $\{x_t\}, t \in [0,1]$. The results can be found at the following link: [https://anonymous.4open.science/r/P-I2SB](https://anonymous.4open.science/r/P-I2SB).
> $$
> forward: X_t\sim q(X_t|X_0, X_1)=\mathcal{N}(X_t;\mu_t(X_0, X_1), \Sigma_t),
> inverse: \hat{X_t} \sim p(X_t|X_0^\epsilon,X_{t+1})=\mathcal{N}(X_t;\mu_\sigma(X_0^{\epsilon}, X_{t+1}), \Sigma_\sigma).
> $$
>
> > Q3. Lines 73 and 147 contain incorrect double quotation marks.
>
> Thank you for pointing this out. Due to compilation issues, we have used **bold** and *italic* formatting instead to avoid errors related to the use of double quotation marks. The revised version is as follows:
> - the ***optimal transport*** between images is not unique
> - a ***relaxed*** version introduced by Kantorovich
>
> > Q4. Line 164 references “problem (2),” but the text does not provide a corresponding explanation.
> - Thank you for your insightful question, which helped us identify an error in Equation (7). The corrected version is as follows, where $\bar{v}(t,x):=f_t(x)-\frac{\sqrt{\beta_t}}{2}\nabla\log\bar{p}(t,x)$ and $p(t,x)\mathrm{d}x = \mu_t(\mathrm{d}x)$.
> $$
> \inf_{(p,v)} \int_0^1\int_{\mathbb{R}^{d}} [\frac{1}{2}||v(t,x)-\bar{v}(t,x)||^2 + \frac{\beta_t}{8}||\nabla\log\frac{p(t,x)}{\bar{p}(t,x)}||^2]p(t,x)\mathrm{d}x\mathrm{d}t. \tag{7}
> $$
> - **The text "as shown in problem (2)" means it is actually the same as equation (2)**. This implies that the SB problem is linked to the optimal transport problem. Furthermore, $p(t,x), (t\in[0,1])$ obtained by solving the SB model represents a form of optimal transport path.
> The phrase "as shown in problem (2)" is incorrect, and we will correct it to "just like equation (2)".
> $$
> \inf_{(\mu,v)} \int_0^1{\int_{\mathbb{R}^{d}} ||v(t,x)||^2 \mu_t(\mathrm{d}x)}\mathrm{d}t. \tag{2}
> $$
>
> > Q5. What is the significance of designing Cat-I2SB? If only the first three channels are retained, equation (12) can be simplified to $(I_{hq}, I_{lq}^d)$.
> - The Cat-I2SB is designed to **implement PHM strategies by hashing training image pairs** from $(I_{hq}, I_{lq}^d)$ to $(I_{hq} \oplus I_{lq}^d, I_{lq}^d \oplus I_{lq}^d)$, enabling the creation of varied training pairs for different degradations $d$.
> - In Equation (12), only the first three channels are used in the final data. However, this hashed data is utilized during P-I2SB training, where the diffusion network's input and output channels are adjusted. The original pairs $(I_{hq}, I_{lq}^d)$ remain unhashed. This modification increases the parameter size of Unet, approximately 558.6MB.

---

> > ### Comment · Reviewer_hmwC · 2025-04-02
> >
> > Thank you to the authors for the detailed rebuttal, which has resolved most of my questions. However, I am still curious about the effect of combining Vanilla-I2SB with data augmentation (such as the data augmentation methods used in DiffBIR) and look forward to further understanding the different impacts of the PHM module and data augmentation methods in the model.

---

> > > ### Author Response · Authors · 2025-04-05
> > >
> > > > Q1. However, I am still curious about the effect of combining Vanilla-I2SB with data augmentation (such as the data augmentation methods used in DiffBIR) and look forward to further understanding the different impacts of the PHM module and data augmentation methods in the model.
> > >
> > > 1. ***Understanding Data Augmentation in DiffBIR***: The restoration process in DiffBIR [1] is divided into two stages: degradation removal (Stage 1) and information regeneration (Stage 2). In Stage 1, DiffBIR employs various restoration modules to address degradations specific to each BIR task. For BFR tasks, DiffBIR utilizes SwinIR as a pre-trained model, allowing it to generalize effectively to unknown degradations by leveraging performance of SwinIR across different tasks.
> > >
> > > 2. ***Combining Vanilla-I2SB with Data Augmentation***: Based on pre-trained degradation removal in Stage 1 of DiffBIR, Vanilla-I2SB can achieve improved performance. This "data augmentation" involves using a lightweight restoration model to initially process low-quality images, reducing the restoration difficulty for Vanilla-I2SB by removing most degradations. However, this approach presents two challenges: first, large parameter size of SwinIR allows it to function as an independent restoration model, potentially skewing fairness when comparing the capabilities of the combined Vanilla-I2SB and SwinIR model with other SOTA methods; second, altering the low-quality input contradicts our exploration of SB models addressing joint distribution issues between complex data distributions, as SwinIR's pre-restoration simplifies the distribution complexity of low-quality images.
> > >
> > > 3. ***Different Impacts*** of the PHM module and data augmentation: **The primary difference lies in PHM targeting high-quality data distributions, while DiffBIR focuses on low-quality ones**. This distinction arises from the different objectives of PHM and DiffBIR. Our PHM applies pseudo-hashing on the high-quality marginal distribution to assist SB models in finding optimal paths, addressing challenges in constructing optimal solutions for blind image restoration tasks. Conversely, DiffBIR uses a pre-trained SwinIR model for initial low-quality image restoration, as non-degraded images serve as better conditions for diffusion-based models, aiming to simplify the detail generation in Stage 2.
> > >
> > > 4. ***Future Directions***: We appreciate your suggestions, which have prompted us to further explore the differences between PHM and SwinIR pre-restoration. We are investigating and experimenting with ways to improve the impact of low- and high-quality marginal distributions on SB models, aiming for better progress in future work.
> > >
> > > _reference: [1]Lin X, He J, Chen Z, et al. DiffBIR: Toward blind image restoration with generative diffusion prior[C]//European Conference on Computer Vision. Cham: Springer Nature Switzerland, 2024: 430-448._

---

### Official Review · Reviewer_TmMa · 2025-03-14

**Overall Recommendation:** 4

**Summary:**

This paper proposes the Pseudo-Hashing Image-to-Image Schrödinger Bridge (P-I2SB), a novel framework for blind face restoration (BFR). The key insight of this paper is that using raw LQ images directly as the starting point for the reverse diffusion process is theoretically optimal.
The authors argue that Schrödinger Bridge (SB)-based approaches offer a better alternative to conventional diffusion-based BFR methods by explicitly learning the optimal transport path between the HQ and LQ distributions. To address limitations in existing SB models (such as optimality and reversibility issues), the paper introduces a Pseudo-Hashing Module (PHM) that preprocesses HQ images into pseudo-samples, ensuring a structurally similar distribution to LQ images. This facilitates an optimal and reversible transformation in the SB framework.
Extensive experiments demonstrate that P-I2SB outperforms prior BFR methods in terms of texture realism, and preservation of facial details.

**Claims And Evidence:**

The authors claim:
1.Schrödinger Bridge-based approaches can offer theoretically optimal transport paths for image restoration.
2.The Pseudo-Hashing Module (PHM) improves SB-based restoration by ensuring reversibility and distribution alignment between LQ and HQ samples.
3.P-I2SB achieves state-of-the-art (SOTA) results in BFR, outperforming existing methods in quality and efficiency.
these claims are supported by theoretical analysis and quantitative results.

**Essential References Not Discussed:**

No

**Experimental Designs Or Analyses:**

The experimental setup is solid

**Methods And Evaluation Criteria:**

The paper presents a strong methodological foundation, leveraging optimal transport theory to redefine BFR as a Schrödinger Bridge problem. The proposed PHM preprocessing step ensures that LQ and HQ distributions align better, improving reversibility in the transformation.

**Other Comments Or Suggestions:**

No

**Other Strengths And Weaknesses:**

Strengths:
(1) Theoretical novelty: Reformulates BFR as an optimal transport problem, leading to a more principled approach.
(2) Practical improvements: Achieves better texture details and inference efficiency than prior methods.
(3) Conceptually elegant: The pseudo-hashing mechanism is a clever solution to the non-reversibility issue in SB models.
Weaknesses:
(1) The computational complexity of PHM preprocessing is not well analyzed.
(2) More ablations on different hashing strategies are required.
(3) Limited real-world evaluations—most experiments use synthetically degraded images.

**Questions For Authors:**

1. Could adaptive pseudo-hashing improve restoration in real-world scenarios?
2.What is the computational cost of PHM preprocessing relative to standard diffusion models?

**Relation To Broader Scientific Literature:**

This work builds upon Diffusion-based image restoration and Schrödinger Bridge methods in generative modeling.

**Theoretical Claims:**

The paper provides a mathematical justification for using Schrödinger Bridge models in BFR. The derivations appear correct, but the paper could include more ablation studies to quantify the impact of PHM on transport efficiency explicitly.

---

> ### Author Rebuttal · Authors · 2025-04-01
>
> Thank you for your detailed feedback and constructive suggestions. Your input is crucial to refining and enhancing the quality of our paper.
> > Q1. The computational complexity of PHM preprocessing is not well analyzed. What is the computational cost of PHM preprocessing relative to standard diffusion models?
>
>  - **Computational Complexity**: In comparison to the baseline methods, our approach incorporates an additional pseudo-hashing module (PHM). The computational complexity of this module is as follows: for Cat-I2SB, it is $\mathcal{O}(n)$; for Res-I2SB, it is $\mathcal{O}(n)$; and for Noise-I2SB, it is $\mathcal{O}(nT)$, where $T=10$ denotes the local number of steps in DDIM, and $n$ denotes the number of input images. This computational complexity is calculated with respect to processing $n$ input images. During inference, our method only adds the Inverse-PHM process relative to the baseline methods, without significantly increasing the complexity of this model or inference time.
>
> > Q2. More ablations on different hashing strategies are required. The paper could include more ablation studies to quantify the impact of PHM on transport efficiencyexplicitly.
>
> - We compared the ablation experiments of different strategies in **Table 3** in main paper and **Figure 7** in the appendix. The related comparison results are provided here again. Additionally, we considered combining all three strategies. Due to time constraints, this was only validated in a toy experiment, as shown in **Sec.4.3 (Toy Exploration and Analysis)**, and more results are shown at the following link: [https://anonymous.4open.science/r/P-I2SB](https://anonymous.4open.science/r/P-I2SB).
>
> - **Table.3 Ablation studies** on CelebA-Test. (a) donates Vanilla-I2SB as the baseline, (b)-(c) compare different condition guidance, and (d)-(f) compare different Pseudo-Hashing strategies. "lq" donates LQ images as the condition in forward and reverse process and "da" donates degradation-aware representation as condition.
>
> | Method          | Condition | Condition | Hashing | Metrics | Metrics | Metrics |
> |-----------------|:---------------:|:---------------:|:--------------------:|:----------------:|:---------------:|:-----------------:|
> |          | lq | da | strategy | NIQE↓ | FID↓ | LPIPS↓ |
> | (a)             |                 |                 |                      | 25.6026          | 6.5709          | 0.2851            |
> | (b)             | ✓               |                 |                      | 25.9522          | 6.4849          | 0.2904            |
> | (c)             | ✓               | ✓               |                      | 25.4932          | 6.3898          | 0.2945            |
> | **(d) ours**    | ✓               | ✓               | Noise                | 18.2646      | 5.7595          | 0.2678            |
> | **(e) ours**    | ✓               | ✓               | Cat                  | 13.9109          | 5.3300      | 0.2395        |
> | **(f) ours**    | ✓               | ✓               | Res                  | 14.2941          | 5.4401          | 0.2431            |
>
> > Q3. Limited real-world evaluations—most experiments use synthetically degraded images. Could adaptive pseudo-hashing improve restoration in real-world scenarios?
>
> - We conducted tests on four datasets in total. Among them, CelebA-Test consists of synthetically degraded images, while ***LFW, CelebChild, WebPhoto-Test, and Wider*** are real-world image datasets. **Table 2** and **Figure 5** in main paper quantitatively and qualitatively compare the restoration performance on these four real-world datasets. Additionally, in **Appendix Sec. J**, we provide a detailed comparison of the restoration effects across the four real-world datasets using Figures 9, 10, 11, and 12.
> - The reason for selecting these four datasets for testing is that previous SOTA methods have been validated on these publicly available datasets. They include various low-quality images with different levels of degradation found in real-world scenarios. This allows for a fairer comparison between our method and the SOTAs.

---

### Official Review · Reviewer_7utM · 2025-03-14

**Overall Recommendation:** 4

**Summary:**

This paper proposes P-I2SB, a novel framework for blind face restoration that leverages a pseudo-hashing strategy to preprocess image pairs and a Schrödinger Bridge Module (SBM) to learn optimal transport paths between LQ and HQ distributions. The key innovation lies in directly using raw LQ images as endpoints in the diffusion process, addressing limitations in existing Schrödinger Bridge (SB) methods related to solution optimality and reversibility. Experiments demonstrate state-of-the-art performance on synthetic and real-world datasets.

## update after rebuttal
Thank you for your reply. After considering the comments from the other reviewers, I have decided to increase the score.

**Claims And Evidence:**

Yes

**Essential References Not Discussed:**

n/a

**Experimental Designs Or Analyses:**

Yes

**Methods And Evaluation Criteria:**

Yes

**Other Comments Or Suggestions:**

n/a

**Other Strengths And Weaknesses:**

Strengths:

1. The pseudo-hashing module (PHM) is a novel and theoretically grounded approach to ensure distribution alignment, enabling direct use of LQ images without complex feature extraction.

2. Comprehensive theoretical analysis justifies the framework’s design and improvements over vanilla SB methods.

Weaknesses:

The paper does not thoroughly analyze the computational overhead of the pseudo-hashing strategies (Cat/Res/Noise-I2SB) compared to baseline methods, despite claiming retained inference speed.

**Questions For Authors:**

- What about the sensitivity to degradation types and the scalability to non-face domains?

- Res-I2SB assumes that the degradation process can be modeled as a linear residual from HQ to LQ, but in reality, degradation (such as JPEG compression and non-uniform blurring) is mostly a nonlinear transformation, which may lead to insufficient fitting ability of the model for complex degradation.

- Cat-I2SB, Res-I2SB, Noise-I2SB still require manual prior selection. I wonder if it would be better to introduce pre-trained degeneration classifiers to guide strategy selection?

**Relation To Broader Scientific Literature:**

n/a

**Theoretical Claims:**

Yes

---

> ### Author Rebuttal · Authors · 2025-04-01
>
> We sincerely appreciate the thorough review and insightful comments you have provided.
> > Q1. The paper does not thoroughly analyze the computational overhead of the pseudo-hashing strategies (Cat/Res/Noise-I2SB) compared to baseline methods, despite claiming retained inference speed.
>
> - **Computational Complexity**: In comparison to the baseline methods, our approach incorporates an additional pseudo-hashing module (PHM). The computational complexity of this module is as follows: for Cat-I2SB, it is $\mathcal{O}(n)$; for Res-I2SB, it is $\mathcal{O}(n)$; and for Noise-I2SB, it is $\mathcal{O}(nT)$, where $T=10$ denotes the local number of steps in DDIM, and $n$ denotes the number of input images. This computational complexity is calculated with respect to processing $n$ input images. During inference, our method only adds the Inverse-PHM process relative to the baseline methods, without significantly increasing the complexity of this model or inference time.
>
> > Q2. What about the sensitivity to degradation types and the scalability to non-face domains?
>
> - **Sensitivity to Degradation Types**: The pseudo-hashing module (PHM) is specifically designed for image restoration under various unknown degradation. The three hashing strategies presented in the paper introduce degradation feature representations either directly or indirectly. This enables the module to perceive both the type and degree of degradation, thereby effectively handling low-quality image restoration across different degradation types.
> - **Scalability to Non-Face Domains**: Our method possesses scalability, and we have verified its effectiveness primarily in blind face restoration (BFR), where the demand for naturalness is exceedingly high. As noted in *"Limitations of Face Image Generation" [1]*, face restoration is more challenging than non-face tasks due to the need for naturalness and identity consistency. The restoration of identity, micro-expressions and other details is crucial, significantly validating the effectiveness of our method. Meanwhile, non-face models emphasize robustness, and currently, no general model can be directly applied to face restoration. We plan to extend our method to general restoration tasks in the future. The advancement of BFR is also critical to the development of face restoration field, and we will continue to advance BFR and expand our research into general domains.
>
> _reference: [1] Rosenberg H, Ahmed S, Ramesh G, et al. Limitations of face image generation. In AAAI, 2024, 38(13): 14838-14846._
> > Q3. Res-I2SB assumes that the degradation process can be modeled as a linear residual from HQ to LQ, but in reality, degradation (such as JPEG compression and non-uniform blurring) is mostly a nonlinear transformation, which may lead to insufficient fitting ability of the model for complex degradation.
>
> - It is instructive for us. The hashing module in Res-I2SB is not designed to construct a linear residual from LQ to HQ. Instead, it aims to model the transformation relationship between two data distributions using a SB-based model. This transformation is often a complex nonlinear one, which necessitates the use of the SB model for resolution. In Res-I2SB, $(I_{lq}, I_{lq} - I_{hq})$ is treated as a new image pair. We do not further explore its linear relationship but require that $I_{lq} - I_{hq}$ serve as a new boundary distribution to help find the optimal path.
> - In this context, the SB model aims to find the optimal joint distribution between given distributions, not to map low-quality to high-quality images. This joint distribution perspective is a broader generalization. Our goal is to establish the optimal dynamic diffusion path between these images, without assuming any specific mapping relationship, such as a linear constraint. Preliminaries (Sec.3) in main paper shows the evolution from a mapping to a joint distribution perspective in optimal transport problems.
>
> > Q4. Cat-I2SB, Res-I2SB, Noise-I2SB still require manual prior selection. I wonder if it would be better to introduce pre-trained degeneration classifiers to guide strategy selection?
>
> - Since we are dealing with a blind task where the degradation parameters and types are random and uncertain, potentially involving multiple combinations, classifiers cannot be used to clearly distinguish each category.
> - In terms of strategy design, Noise-I2SB handles differences caused by various degradations using degradation representation. We employ the formula $I_{hq}^{noise} = I_{lq}^d + \lambda_d \epsilon$, where $\lambda_d$ represents the noise level, which is directly related to $I_{lq}^d$. Here, we set $\lambda_d$ as the degeneration representation feature of $I_{lq}^d$ with an MLP layer.
> - Looking forward, we will focus on researching how to design PHM strategies without relying on manual prior selection. This will require further exploration of the relationship between the conditions for the optimal solution in SB and the boundary constraints.

---

### Decision · Program_Chairs · 2025-05-01

**Decision:**

Accept (poster)

**Comment:**

This paper proposes P-I2SB, a novel blind face restoration framework based on Schrödinger Bridge and inspired by optimal transport theory. By hashing HQ images into pseudo-samples aligned with LQ distributions, it ensures optimal and reversible restoration, achieving state-of-the-art results with better textures and efficiency. The paper receives all positive reviews, with one reviewer upgrading their score from weak accept to accept after the rebuttal. Based on the overall enthusiasm and the strengthened justification provided during the rebuttal, the paper is recommended for acceptance. The authors are encouraged to incorporate the clarifications and discussions from the rebuttal into the final version.